# On the Global Convergence of (Fast) Incremental Expectation Maximization Methods

**Belhal Karimi**
CMAP, École Polytechnique
Palaiseau, France
belhal.karimi@polytechnique.edu

**Hoi-To Wai**
The Chinese University of Hong Kong
Shatin, Hong Kong
htwai@se.cuhk.edu.hk

**Eric Moulines**
CMAP, École Polytechnique
Palaiseau, France
eric.moulines@polytechnique.edu

**Marc Lavielle**
INRIA Saclay
Palaiseau, France
marc.lavielle@inria.fr

## Abstract

The EM algorithm is one of the most popular algorithm for inference in latent data models. The original formulation of the EM algorithm does not scale to large data set, because the whole data set is required at each iteration of the algorithm. To alleviate this problem, Neal and Hinton [1998] have proposed an incremental version of the EM (iEM) in which at each iteration the conditional expectation of the latent data (E-step) is updated only for a mini-batch of observations. Another approach has been proposed by Cappé and Moulines [2009] in which the E-step is replaced by a stochastic approximation step, closely related to stochastic gradient. In this paper, we analyze incremental and stochastic version of the EM algorithm as well as the variance reduced-version of [Chen et al., 2018] in a common unifying framework. We also introduce a new version incremental version, inspired by the SAGA algorithm by Defazio et al. [2014]. We establish non-asymptotic convergence bounds for global convergence. Numerical applications are presented in this article to illustrate our findings.

## 1 Introduction

Many problems in machine learning pertain to tackling an empirical risk minimization of the form

$$\min_{\boldsymbol{\theta} \in \Theta} \overline{\mathcal{L}}(\boldsymbol{\theta}) := \mathrm{R}(\boldsymbol{\theta}) + \mathcal{L}(\boldsymbol{\theta}) \text{ with } \mathcal{L}(\boldsymbol{\theta}) = \frac{1}{n} \sum_{i=1}^{n} \mathcal{L}_i(\boldsymbol{\theta}) := \frac{1}{n} \sum_{i=1}^{n} \big\{ -\log g(y_i; \boldsymbol{\theta}) \big\} , \quad (1)$$

where $\{y_i\}_{i=1}^{n}$ are the observations, $\Theta$ is a convex subset of $\mathbb{R}^d$ for the parameters, $\mathrm{R} : \Theta \to \mathbb{R}$ is a smooth convex regularization function and for each $\boldsymbol{\theta} \in \Theta$, $g(y; \boldsymbol{\theta})$ is the (incomplete) likelihood of each individual observation. The objective function $\overline{\mathcal{L}}(\boldsymbol{\theta})$ is possibly *non-convex* and is assumed to be lower bounded $\overline{\mathcal{L}}(\boldsymbol{\theta}) > -\infty$ for all $\boldsymbol{\theta} \in \Theta$. In the latent variable model, $g(y_i; \boldsymbol{\theta})$, is the marginal of the complete data likelihood defined as $f(z_i, y_i; \boldsymbol{\theta})$, i.e. $g(y_i; \boldsymbol{\theta}) = \int_{\mathsf{Z}} f(z_i, y_i; \boldsymbol{\theta}) \mu(\mathrm{d}z_i)$, where $\{z_i\}_{i=1}^{n}$ are the (unobserved) latent variables. We consider the setting where the complete data likelihood belongs to the curved exponential family, *i.e.,*

$$f(z_i, y_i; \boldsymbol{\theta}) = h(z_i, y_i) \exp \big( \langle S(z_i, y_i) \, | \, \phi(\boldsymbol{\theta}) \rangle - \psi(\boldsymbol{\theta}) \big) , \quad (2)$$

where $\psi(\boldsymbol{\theta})$, $h(z_i, y_i)$ are scalar functions, $\phi(\boldsymbol{\theta}) \in \mathbb{R}^k$ is a vector function, and $S(z_i, y_i) \in \mathbb{R}^k$ is the complete data sufficient statistics. Latent variable models are widely used in machine learning and

statistics; examples include mixture models for density estimation, clustering document, and topic modelling; see [McLachlan and Krishnan, 2007] and the references therein.

The basic "batch" EM (bEM) method iteratively computes a sequence of estimates $\{\boldsymbol{\theta}^k, k \in \mathbb{N}\}$ with an initial parameter $\boldsymbol{\theta}^0$. Each iteration of bEM is composed of two steps. In the E-step, a surrogate function is computed as $\boldsymbol{\theta} \mapsto Q(\boldsymbol{\theta}, \boldsymbol{\theta}^{k-1}) = \sum_{i=1}^{n} Q_i(\boldsymbol{\theta}, \boldsymbol{\theta}^{k-1})$ where $Q_i(\boldsymbol{\theta}, \boldsymbol{\theta}') :=$ $-\int_{\mathsf{Z}} \log f(z_i, y_i; \theta) p(z_i|y_i; \boldsymbol{\theta}') \mu(\mathrm{d}z_i)$ such that $p(z_i|y_i; \boldsymbol{\theta}) := f(z_i, y_i; \boldsymbol{\theta})/g(y_i, \boldsymbol{\theta})$ is the conditional probability density of the latent variables $z_i$ given the observations $y_i$. When $f(z_i, y_i; \boldsymbol{\theta})$ follows the curved exponential family model, the E-step amounts to computing the conditional expectation of the complete data sufficient statistics,

$$\bar{\mathbf{s}}(\boldsymbol{\theta}) = \frac{1}{n} \sum_{i=1}^{n} \bar{\mathbf{s}}_i(\boldsymbol{\theta}) \quad \text{where} \quad \bar{\mathbf{s}}_i(\boldsymbol{\theta}) = \int_{\mathsf{Z}} S(z_i, y_i) p(z_i|y_i; \boldsymbol{\theta}) \mu(\mathrm{d}z_i). \quad (3)$$

In the M-step, the surrogate function is minimized producing a new fit of the parameter $\boldsymbol{\theta}^k = \arg\max_{\boldsymbol{\theta} \in \Theta} Q(\boldsymbol{\theta}, \boldsymbol{\theta}^{k-1})$. The EM method has several appealing features – it is monotone where the likelihood do not decrease at each iteration, invariant with respect to the parameterization, numerically stable when the optimization set is well defined, etc. The EM method has been the subject of considerable interest since its formalization in [Dempster et al., 1977].

With the sheer size of data sets today, the bEM method is not applicable as the E-step (3) involves a full pass over the dataset of $n$ observations. Several approaches based on stochastic optimization have been proposed to address this problem. Neal and Hinton [1998] proposed (but not analyzed) an incremental version of EM, referred to as the iEM method. Cappé and Moulines [2009] developed the online EM (sEM) method which uses a stochastic approximation procedure to track the sufficient statistics defined in (3). Recently, Chen et al. [2018] proposed a variance reduced sEM (sEM-VR) method which is inspired by the SVRG algorithm popular in stochastic convex optimization [Johnson and Zhang, 2013]. The applications of the above stochastic EM methods are numerous, especially with the iEM and sEM methods; e.g., [Thiesson et al., 2001] for inference with missing data, [Ng and McLachlan, 2003] for mixture models and unsupervised clustering, [Hinton et al., 2006] for inference of deep belief networks, [Hofmann, 1999] for probabilistic latent semantic analysis, [Wainwright et al., 2008, Blei et al., 2017] for variational inference of graphical models and [Ablin et al., 2018] for Independent Component Analysis.

This paper focuses on the theoretical aspect of stochastic EM methods by establishing novel *non-asymptotic* and *global* convergence rates for them. Our contributions are as follows.

- We offer two complementary views for the global convergence of EM methods – one focuses on the parameter space, and one on the sufficient statistics space. On one hand, the EM method can be studied as an *majorization-minimization* (MM) method in the parameter space. On the other hand, the EM method can be studied as a *scaled-gradient method* in the sufficient statistics space.

- Based on the two views described, we derive non-asymptotic convergence rate for stochastic EM methods. First, we show that the iEM method [Neal and Hinton, 1998] is a special instance of the MISO framework [Mairal, 2015], and takes $\mathcal{O}(n/\epsilon)$ iterations to find an $\epsilon$-stationary point to the ML estimation problem. Second, the sEM-VR method [Chen et al., 2018] is an instance of variance reduced stochastic scaled-gradient method, which takes $\mathcal{O}(n^{2/3}/\epsilon)$ iterations to find to an $\epsilon$-stationary point.

- Lastly, we develop a Fast Incremental EM (fiEM) method based on the SAGA algorithm [Defazio et al., 2014, Reddi et al., 2016b] for stochastic optimization. We show that the new method is again a scaled-gradient method with the same iteration complexity as sEM-VR. This new method offers trade-off between storage cost and computation complexity.

Importantly, our results capitalizes on the efficiency of stochastic EM methods applied on large datasets, and we support the above findings using numerical experiments.

**Prior Work** Since the empirical risk minimization problem (1) is typically *non-convex*, most prior work studying the convergence of EM methods considered either the *asymptotic* and/or *local* behaviors. For the classical study, the global convergence to a stationary point (either a local minimum or a saddle point) of the bEM method has been established by Wu et al. [1983] (by making the arguments

developed in Dempster et al. [1977] rigorous). The global convergence is a direct consequence of the EM method to be monotone. It is also known that in the neighborhood of a stationary point and under regularity conditions, the local rate of convergence of the bEM is linear and is given by the amount of *missing information* [McLachlan and Krishnan, 2007, Chapters 3 and 4].

The convergence of the iEM method was first tackled by Gunawardana and Byrne [2005] exploiting the interpretation of the method as an alternating minimization procedure under the information geometric framework developed in [Csiszár and Tusnády, 1984]. Although the EM algorithm is presented as an alternation between the E-step and M-step, it is also possible to take a variational perspective on EM to view both steps as maximization steps. Nevertheless, Gunawardana and Byrne [2005] assume that the latent variables take only a finite number of values and the order in which the observations are processed remains the same from one pass to the other.

More recently, the *local but non-asymptotic convergence* of EM methods has been studied in several works. These results typically require the initializations to be within a neighborhood of an isolated stationary point and the (negated) log-likelihood function to be strongly convex locally. Such conditions are either difficult to verify in general or have been derived only for specific models; see for example [Wang et al., 2015, Xu et al., 2016, Balakrishnan et al., 2017] and the references therein. The local convergence of sEM-VR method has been studied in [Chen et al., 2018, Theorem 1] but under a pathwise global stability condition. The authors' work [Karimi et al., 2019] provided the first global non-asymptotic analysis of the online (stochastic) EM method [Cappé and Moulines, 2009]. In comparison, the present work analyzes the variance reduced variants of EM method. Lastly, it is worthwhile to mention that Zhu et al. [2017] analyzed a variance reduced *gradient* EM method similar to [Balakrishnan et al., 2017].

## 2 Stochastic Optimization Techniques for EM methods

Let $k \geq 0$ be the iteration number. The $k$th iteration of a generic stochastic EM method is composed of two sub-steps — firstly,

$$\textsf{sE-step}: \quad \hat{\mathbf{s}}^{(k+1)} = \hat{\mathbf{s}}^{(k)} - \gamma_{k+1}\big(\hat{\mathbf{s}}^{(k)} - \boldsymbol{\mathcal{S}}^{(k+1)}\big), \tag{4}$$

which is a stochastic version of the E-step in (3). Note $\{\gamma_k\}_{k=1}^{\infty} \in [0,1]$ is a sequence of step sizes, $\boldsymbol{\mathcal{S}}^{(k+1)}$ is a proxy for $\overline{\mathbf{s}}(\hat{\boldsymbol{\theta}}^{(k)})$, and $\overline{\mathbf{s}}$ is defined in (3). Secondly, the M-step is given by

$$\textsf{M-step}: \quad \hat{\boldsymbol{\theta}}^{(k+1)} = \overline{\boldsymbol{\theta}}(\hat{\boldsymbol{s}}^{(k+1)}) := \underset{\boldsymbol{\theta} \in \Theta}{\arg\min} \ \big\{ \mathrm{R}(\boldsymbol{\theta}) + \psi(\boldsymbol{\theta}) - \big\langle \hat{\boldsymbol{s}}^{(k+1)} \,|\, \phi(\boldsymbol{\theta}) \big\rangle \big\}, \tag{5}$$

which depends on the sufficient statistics in the sE-step. The stochastic EM methods differ in the way that $\boldsymbol{\mathcal{S}}^{(k+1)}$ is computed. Existing methods employ stochastic approximation or variance reduction without the need to fully compute $\overline{\mathbf{s}}(\hat{\boldsymbol{\theta}}^{(k)})$. To simplify notations, we define

$$\overline{\mathbf{s}}_i^{(k)} := \overline{\mathbf{s}}_i(\hat{\boldsymbol{\theta}}^{(k)}) = \int_{\mathsf{Z}} S(z_i, y_i) p(z_i|y_i; \hat{\boldsymbol{\theta}}^{(k)}) \mu(\mathrm{d}z_i) \quad \text{and} \quad \overline{\mathbf{s}}^{(\ell)} := \overline{\mathbf{s}}(\hat{\boldsymbol{\theta}}^{(\ell)}) = \frac{1}{n}\sum_{i=1}^n \overline{\mathbf{s}}_i^{(\ell)}. \tag{6}$$

If $\boldsymbol{\mathcal{S}}^{(k+1)} = \overline{\mathbf{s}}^{(k)}$ and $\gamma_{k+1} = 1$, (4) reduces to the E-step in the classical bEM method. To formally describe the stochastic EM methods, we let $i_k \in [\![1, n]\!]$ be a random index drawn at iteration $k$ and $\tau_i^k = \max\{k' : i_{k'} = i, \ k' < k\}$ be the iteration index such that $i \in [\![1, n]\!]$ is last drawn prior to iteration $k$. The proxy $\boldsymbol{\mathcal{S}}^{(k+1)}$ in (4) is drawn as:

$$\textit{(iEM [Neal and Hinton, 1998])} \qquad \boldsymbol{\mathcal{S}}^{(k+1)} = \boldsymbol{\mathcal{S}}^{(k)} + \tfrac{1}{n}\big(\overline{\mathbf{s}}_{i_k}^{(k)} - \overline{\mathbf{s}}_{i_k}^{(\tau_{i_k}^k)}\big) \tag{7}$$

$$\textit{(sEM [Cappé and Moulines, 2009])} \qquad \boldsymbol{\mathcal{S}}^{(k+1)} = \overline{\mathbf{s}}_{i_k}^{(k)} \tag{8}$$

$$\textit{(sEM-VR [Chen et al., 2018])} \qquad \boldsymbol{\mathcal{S}}^{(k+1)} = \overline{\mathbf{s}}^{(\ell(k))} + \big(\overline{\mathbf{s}}_{i_k}^{(k)} - \overline{\mathbf{s}}_{i_k}^{(\ell(k))}\big) \tag{9}$$

The stepsize is set to $\gamma_{k+1} = 1$ for the iEM method; $\gamma_{k+1} = \gamma$ is constant for the sEM-VR method. In the original version of the sEM method, the sequence of step $\gamma_{k+1}$ is a diminishing step size. Moreover, for iEM we initialize with $\boldsymbol{\mathcal{S}}^{(0)} = \overline{\mathbf{s}}^{(0)}$; for sEM-VR, we set an epoch size of $m$ and define $\ell(k) := m\lfloor k/m \rfloor$ as the first iteration number in the epoch that iteration $k$ is in.

**fiEM** Our analysis framework can handle a new, yet natural application of a popular variance reduction technique to the EM method. The new method, called fiEM, is developed from the SAGA method [Defazio et al., 2014] in a similar vein as in sEM-VR.

For iteration $k \geq 0$, the fiEM method draws *two* indices *independently* and uniformly as $i_k, j_k \in [\![1, n]\!]$. In addition to $\tau_i^k$ which was defined *w.r.t.* $i_k$, we define $t_j^k = \{k' : j_{k'} = j, k' < k\}$ to be the iteration index where the sample $j \in [\![1, n]\!]$ is last drawn as $j_k$ prior to iteration $k$. With the initialization $\overline{\boldsymbol{S}}^{(0)} = \overline{\mathbf{s}}^{(0)}$, we use a slightly different update rule from SAGA inspired by [Reddi et al., 2016b], as described by the following recursive updates

$$\boldsymbol{S}^{(k+1)} = \overline{\boldsymbol{S}}^{(k)} + \big(\overline{\mathbf{s}}_{i_k}^{(k)} - \overline{\mathbf{s}}_{i_k}^{(t_{i_k}^k)}\big), \quad \overline{\boldsymbol{S}}^{(k+1)} = \overline{\boldsymbol{S}}^{(k)} + n^{-1}\big(\overline{\mathbf{s}}_{j_k}^{(k)} - \overline{\mathbf{s}}_{j_k}^{(t_{j_k}^k)}\big). \tag{10}$$

where we set a constant step size as $\gamma_{k+1} = \gamma$.

In the above, the update of $\boldsymbol{S}^{(k+1)}$ corresponds to an *unbiased estimate* of $\overline{\mathbf{s}}^{(k)}$, while the update for $\overline{\boldsymbol{S}}^{(k+1)}$ maintains the structure that $\overline{\boldsymbol{S}}^{(k)} = n^{-1}\sum_{i=1}^{n} \overline{\mathbf{s}}_i^{(t_i^k)}$ for any $k \geq 0$. The two updates of (10) are based on two different and independent indices $i_k, j_k$ that are randomly drawn from $[\![n]\!]$. This is used for our fast convergence analysis in Section 3.

We summarize the iEM, sEM-VR, sEM, fiEM methods in Algorithm 1. The random termination number (11) is inspired by [Ghadimi and Lan, 2013] which enables one to show non-asymptotic convergence to stationary point for non-convex optimization. Due to their stochastic nature, the per-iteration complexity for all the stochastic EM methods are independent of $n$, unlike the bEM method. They are thus applicable to large datasets with $n \gg 1$.

---

**Algorithm 1** Stochastic EM methods.

1: **Input:** initializations $\hat{\boldsymbol{\theta}}^{(0)} \leftarrow 0$, $\hat{\mathbf{s}}^{(0)} \leftarrow \overline{\mathbf{s}}^{(0)}$, $K_{\mathsf{max}} \leftarrow$ max. iteration number.
2: Set the terminating iteration number, $K \in \{0, \ldots, K_{\mathsf{max}} - 1\}$, as a discrete r.v. with:

$$P(K = k) = \frac{\gamma_k}{\sum_{\ell=0}^{K_{\mathsf{max}}-1} \gamma_\ell}. \tag{11}$$

3: **for** $k = 0, 1, 2, \ldots, K$ **do**
4:     Draw index $i_k \in [\![1, n]\!]$ uniformly (and $j_k \in [\![1, n]\!]$ for fiEM).
5:     Compute the surrogate sufficient statistics $\boldsymbol{S}^{(k+1)}$ using (8) or (7) or (9) or (10).
6:     Compute $\hat{\mathbf{s}}^{(k+1)}$ via the **sE-step** (4).
7:     Compute $\hat{\boldsymbol{\theta}}^{(k+1)}$ via the **M-step** (5).
8: **end for**
9: **Return**: $\hat{\boldsymbol{\theta}}^{(K)}$.

---

### 2.1 Example: Gaussian Mixture Model

We discuss an example of learning a Gaussian Mixture Model (GMM) from a set of $n$ observations $\{y_i\}_{i=1}^n$. We focus on a simplified setting where there are $M$ components of unit variance and unknown means, the GMM is parameterized by $\boldsymbol{\theta} = (\{\omega_m\}_{m=1}^{M-1}, \{\mu_m\}_{m=1}^{M}) \in \Theta = \Delta^M \times \mathbb{R}^M$, where $\Delta^M \subseteq \mathbb{R}^{M-1}$ is the reduced $M$-dimensional probability simplex [see (29)]. We use the penalization $\mathrm{R}(\boldsymbol{\theta}) = \frac{\delta}{2}\sum_{m=1}^{M}\mu_m^2 - \log \mathrm{Dir}(\boldsymbol{\omega}; M, \epsilon)$ where $\delta > 0$ and $\mathrm{Dir}(\cdot; M, \epsilon)$ is the $M$ dimensional symmetric Dirichlet distribution with concentration parameter $\epsilon > 0$. Furthermore, we use $z_i \in [\![M]\!]$ as the latent label. The complete data log-likelihood is given by

$$\log f(z_i, y_i; \boldsymbol{\theta}) = \sum_{m=1}^{M} \mathbb{1}_{\{m=z_i\}}\big[\log(\omega_m) - \mu_m^2/2\big] + \sum_{m=1}^{M} \mathbb{1}_{\{m=z_i\}}\mu_m y_i + \mathrm{constant}, \tag{12}$$

where $\mathbb{1}_{\{m=z_i\}} = 1$ if $m = z_i$; otherwise $\mathbb{1}_{\{m=z_i\}} = 0$. The above can be rewritten in the same form as (2), particularly with $S(y_i, z_i) \equiv (s_{i,1}^{(1)}, ..., s_{i,M-1}^{(1)}, s_{i,1}^{(2)}, ..., s_{i,M-1}^{(2)}, s_i^{(3)})$ and $\phi(\boldsymbol{\theta}) \equiv (\phi_1^{(1)}(\boldsymbol{\theta}), ..., \phi_{M-1}^{(1)}(\boldsymbol{\theta}), \phi_1^{(2)}(\boldsymbol{\theta}), ..., \phi_{M-1}^{(2)}(\boldsymbol{\theta}), \phi^{(3)}(\boldsymbol{\theta}))$ such that

$$\begin{aligned}
s_{i,m}^{(1)} &= \mathbb{1}_{\{z_i=m\}}, \quad \phi_m^{(1)}(\boldsymbol{\theta}) = \{\log(\omega_m) - \mu_m^2/2\} - \{\log(1 - \textstyle\sum_{j=1}^{M-1}\omega_j) - \mu_M^2/2\}, \\
s_{i,m}^{(2)} &= \mathbb{1}_{\{z_i=m\}}y_i, \quad \phi_m^{(2)}(\boldsymbol{\theta}) = \mu_m, \quad s_i^{(3)} = y_i, \quad \phi^{(3)}(\boldsymbol{\theta}) = \mu_M,
\end{aligned} \tag{13}$$

and $\psi(\boldsymbol{\theta}) = -\{\log(1 - \sum_{m=1}^{M-1}\omega_m) - \mu_M^2/2\sigma^2\}$. To evaluate the **sE-step**, the conditional expectation required by (6) can be computed in closed form, as they depend on $\mathbb{E}_{\hat{\boldsymbol{\theta}}^{(k)}}[\mathbb{1}_{\{z_i=m\}}|y = y_i]$ and $\mathbb{E}_{\hat{\boldsymbol{\theta}}^{(k)}}[y_i \mathbb{1}_{\{z_i=m\}}|y = y_i]$. Moreover, the **M-step** (5) solves a strongly convex problem and can

be computed in closed form. Given a sufficient statistics $s \equiv (s^{(1)}, s^{(2)}, s^{(3)})$, the solution to (5) is:

$$\overline{\boldsymbol{\theta}}(s) = \begin{pmatrix} (1 + \epsilon M)^{-1}\big(s_1^{(1)} + \epsilon, \dots, s_{M-1}^{(1)} + \epsilon\big)^\top \\ \big((s_1^{(1)} + \delta)^{-1}s_1^{(2)}, \dots, (s_{M-1}^{(1)} + \delta)^{-1}s_{M-1}^{(2)}\big)^\top \\ \big(1 - \sum_{m=1}^{M-1} s_m^{(1)} + \delta\big)^{-1}\big(s^{(3)} - \sum_{m=1}^{M-1} s_m^{(2)}\big) \end{pmatrix}. \tag{14}$$

The next section presents the main results of this paper for the convergence of stochastic EM methods. We shall use the above example on GMM to illustrate the required assumptions.

## 3 Global Convergence of Stochastic EM Methods

We establish non-asymptotic rates for the *global convergence* of the stochastic EM methods. We show that the iEM method is an instance of the incremental MM method; while sEM-VR, fiEM methods are instances of variance reduced *stochastic scaled gradient* methods. As we will see, the latter interpretation allows us to establish fast convergence rates of sEM-VR and fiEM methods. Detailed proofs for the theoretical results in this section are relegated to the appendix.

First, we list a few assumptions which will enable the convergence analysis performed later in this section. Define:

$$\mathsf{S} := \Big\{ \sum_{i=1}^n \alpha_i \mathbf{s}_i \; : \; \mathbf{s}_i \in \mathrm{conv}\left\{ S(z, y_i) \; : \; z \in \mathsf{Z} \right\}, \; \alpha_i \in [-1, 1], \; i \in [\![1, n]\!] \Big\}, \tag{15}$$

where $\mathrm{conv}\{A\}$ denotes the closed convex hull of the set $A$. From (15), we observe that the iEM, sEM-VR, and fiEM methods generate $\hat{s}^{(k)} \in \mathsf{S}$ for any $k \geq 0$. Consider:

**H1.** *The sets* $\mathsf{Z}, \mathsf{S}$ *are compact. There exists constants* $C_\mathsf{S}, C_\mathsf{Z}$ *such that:*

$$C_\mathsf{S} := \max_{\mathbf{s}, \mathbf{s}' \in \mathsf{S}} \|\mathbf{s} - \mathbf{s}'\| < \infty, \quad C_\mathsf{Z} := \max_{i \in [\![1,n]\!]} \int_\mathsf{Z} |S(z, y_i)| \mu(\mathrm{d}z) < \infty. \tag{16}$$

H1 depends on the latent data model used and can be satisfied by several practical models. For instance, the GMM in Section 2.1 satisfies (16) as the sufficient statistics are composed of indicator functions and observations. Other examples can also be found in Section 4. Denote by $\mathrm{J}_\kappa^{\boldsymbol{\theta}}(\boldsymbol{\theta}')$ the Jacobian of the function $\kappa : \boldsymbol{\theta} \mapsto \kappa(\boldsymbol{\theta})$ at $\boldsymbol{\theta}' \in \Theta$. Consider:

**H2.** *The function* $\phi$ *is smooth and bounded on* $\mathrm{int}(\Theta)$, *i.e., the interior of* $\Theta$. *For all* $\boldsymbol{\theta}, \boldsymbol{\theta}' \in \mathrm{int}(\Theta)^2$, $\| \mathrm{J}_\phi^{\boldsymbol{\theta}}(\boldsymbol{\theta}) - \mathrm{J}_\phi^{\boldsymbol{\theta}}(\boldsymbol{\theta}')\| \leq \mathrm{L}_\phi \|\boldsymbol{\theta} - \boldsymbol{\theta}'\|$ *and* $\| \mathrm{J}_\phi^{\boldsymbol{\theta}}(\boldsymbol{\theta}')\| \leq C_\phi$.

**H3.** *The conditional distribution is smooth on* $\mathrm{int}(\Theta)$. *For any* $i \in [\![1, n]\!]$, $z \in \mathsf{Z}$, $\boldsymbol{\theta}, \boldsymbol{\theta}' \in \mathrm{int}(\Theta)^2$, *we have* $\big|p(z|y_i; \boldsymbol{\theta}) - p(z|y_i; \boldsymbol{\theta}')\big| \leq \mathrm{L}_p \|\boldsymbol{\theta} - \boldsymbol{\theta}'\|$.

**H4.** *For any* $s \in \mathsf{S}$, *the function* $\boldsymbol{\theta} \mapsto L(s, \boldsymbol{\theta}) := \mathrm{R}(\boldsymbol{\theta}) + \psi(\boldsymbol{\theta}) - \langle \mathbf{s} \,|\, \phi(\boldsymbol{\theta}) \rangle$ *admits a unique global minimum* $\overline{\boldsymbol{\theta}}(\mathbf{s}) \in \mathrm{int}(\Theta)$. *In addition,* $\mathrm{J}_\phi^{\boldsymbol{\theta}}(\overline{\boldsymbol{\theta}}(\mathbf{s}))$ *is full rank and* $\overline{\boldsymbol{\theta}}(\mathbf{s})$ *is* $\mathrm{L}_\theta$-*Lipschitz*.

Under H1, the assumptions H2 and H3 are standard for the curved exponential family distribution and the conditional probability distributions, respectively; H4 can be enforced by designing a strongly convex regularization function $\mathrm{R}(\boldsymbol{\theta})$ tailor made for $\Theta$. For instance, the penalization for GMM in Section 2.1 ensures $\boldsymbol{\theta}^{(k)}$ is unique and lies in $\mathrm{int}(\Delta^M) \times \mathbb{R}^M$, which can further imply the second statement in H4. We remark that for H3, it is possible to define the Lipschitz constant $\mathrm{L}_p$ independently for each data $y_i$ to yield a refined characterization. We did not pursue such assumption to keep the notations simple.

Denote by $\mathrm{H}_L^{\boldsymbol{\theta}}(\mathbf{s}, \boldsymbol{\theta})$ the Hessian w.r.t to $\boldsymbol{\theta}$ for a given value of $\mathbf{s}$ of the function $\boldsymbol{\theta} \mapsto L(\mathbf{s}, \boldsymbol{\theta}) = \mathrm{R}(\boldsymbol{\theta}) + \psi(\boldsymbol{\theta}) - \langle \mathbf{s} \,|\, \phi(\boldsymbol{\theta}) \rangle$, and define

$$\mathrm{B}(\mathbf{s}) := \mathrm{J}_\phi^{\boldsymbol{\theta}}(\overline{\boldsymbol{\theta}}(\mathbf{s})) \Big( \mathrm{H}_L^{\boldsymbol{\theta}}(\mathbf{s}, \overline{\boldsymbol{\theta}}(\mathbf{s})) \Big)^{-1} \mathrm{J}_\phi^{\boldsymbol{\theta}}(\overline{\boldsymbol{\theta}}(\mathbf{s}))^\top. \tag{17}$$

**H5.** *It holds that* $v_{\max} := \sup_{\mathbf{s} \in \mathsf{S}} \|\mathrm{B}(\mathbf{s})\| < \infty$ *and* $0 < v_{\min} := \inf_{\mathbf{s} \in \mathsf{S}} \lambda_{\min}(\mathrm{B}(\mathbf{s}))$. *There exists a constant* $\mathrm{L}_B$ *such that for all* $\mathbf{s}, \mathbf{s}' \in \mathsf{S}^2$, *we have* $\|\mathrm{B}(\mathbf{s}) - \mathrm{B}(\mathbf{s}')\| \leq \mathrm{L}_B \|\mathbf{s} - \mathbf{s}'\|$.

Again, H5 is satisfied by practical models. For GMM in Section 2.1, it can be verified by deriving the closed form expression for $\mathrm{B}(\mathbf{s})$ and using H1; also see the other example in Section 4. The derivation is, however, technical and will be relegated to the supplementary material.

Under H1, we have $\|\hat{s}^{(k)}\| < \infty$ since $\mathsf{S}$ is compact. On the other hand, under H4, the EM methods generate $\hat{\boldsymbol{\theta}}^{(k)} \in \mathrm{int}(\Theta)$ for any $k \geq 0$. Overall, these assumptions ensure that the EM methods operate in a 'nice' set throughout the optimization process.

## 3.1 Incremental EM method

We show that the iEM method is a special case of the MISO method [Mairal, 2015] utilizing the majorization minimization (MM) technique. The latter is a common technique for handling non-convex optimization. We begin by defining a surrogate function that majorizes $\mathcal{L}_i$:

$$Q_i(\boldsymbol{\theta}; \boldsymbol{\theta}') := - \int_{\mathsf{Z}} \{\log f(z_i, y_i; \boldsymbol{\theta}) - \log p(z_i | y_i; \boldsymbol{\theta}')\} \, p(z_i | y_i; \boldsymbol{\theta}') \mu(\mathrm{d}z_i) . \tag{18}$$

The second term inside the bracket is a constant that does not depend on the first argument $\boldsymbol{\theta}$. Since $f(z_i, y_i; \boldsymbol{\theta}) = p(z_i | y_i; \boldsymbol{\theta}) g(y_i; \boldsymbol{\theta})$, for all $\boldsymbol{\theta}' \in \Theta$, we get $Q_i(\boldsymbol{\theta}'; \boldsymbol{\theta}') = -\log g(y_i; \boldsymbol{\theta}') = \mathcal{L}_i(\boldsymbol{\theta}')$. For all $\boldsymbol{\theta}, \boldsymbol{\theta}' \in \Theta$, applying the Jensen inequality shows

$$Q_i(\boldsymbol{\theta}, \boldsymbol{\theta}') - \mathcal{L}_i(\boldsymbol{\theta}) = \int \log \frac{p(z_i | y_i; \boldsymbol{\theta}')}{p(z_i | y_i; \boldsymbol{\theta})} p(z_i | y_i; \boldsymbol{\theta}') \mu(\mathrm{d}z_i) \geq 0 \tag{19}$$

which is the Kullback-Leibler divergence between the conditional distribution of the latent data $p(\cdot | y_i; \boldsymbol{\theta})$ and $p(\cdot | y_i; \boldsymbol{\theta}')$. Hence, for all $i \in [\![1, n]\!]$, $Q_i(\boldsymbol{\theta}; \boldsymbol{\theta}')$ is a majorizing surrogate to $\mathcal{L}_i(\boldsymbol{\theta})$, *i.e.*, it satisfies for all $\boldsymbol{\theta}, \boldsymbol{\theta}' \in \Theta$, $Q_i(\boldsymbol{\theta}; \boldsymbol{\theta}') \geq \mathcal{L}_i(\boldsymbol{\theta})$ with equality when $\boldsymbol{\theta} = \boldsymbol{\theta}'$. For the special case of curved exponential family distribution, the M-step of the iEM method is expressed as

$$\begin{aligned} \hat{\boldsymbol{\theta}}^{(k+1)} &\in \arg\min_{\boldsymbol{\theta} \in \Theta} \left\{ \mathrm{R}(\boldsymbol{\theta}) + n^{-1} \sum_{i=1}^{n} Q_i(\boldsymbol{\theta}; \hat{\boldsymbol{\theta}}^{(\tau_i^{(k+1)})}) \right\} \\ &= \arg\min_{\boldsymbol{\theta} \in \Theta} \left\{ \mathrm{R}(\boldsymbol{\theta}) + \psi(\boldsymbol{\theta}) - \left\langle n^{-1} \sum_{i=1}^{n} \bar{\mathbf{s}}_i^{(\tau_i^{k+1})} \,|\, \phi(\boldsymbol{\theta}) \right\rangle \right\}. \end{aligned} \tag{20}$$

The iEM method can be interpreted through the MM technique — in the M-step, $\hat{\boldsymbol{\theta}}^{(k+1)}$ minimizes an upper bound of $\overline{\mathcal{L}}(\boldsymbol{\theta})$, while the sE-step updates the surrogate function in (20) which tightens the upper bound. Importantly, the error between the surrogate function and $\mathcal{L}_i$ is a smooth function:

**Lemma 1.** *Assume H1, H2, H3, H4. Let $e_i(\boldsymbol{\theta}; \boldsymbol{\theta}') := Q_i(\boldsymbol{\theta}; \boldsymbol{\theta}') - \mathcal{L}_i(\boldsymbol{\theta})$. For any $\boldsymbol{\theta}, \bar{\boldsymbol{\theta}}, \boldsymbol{\theta}' \in \Theta^3$, we have $\|\nabla e_i(\boldsymbol{\theta}; \boldsymbol{\theta}') - \nabla e_i(\bar{\boldsymbol{\theta}}; \boldsymbol{\theta}')\| \leq \mathrm{L}_e \|\boldsymbol{\theta} - \bar{\boldsymbol{\theta}}\|$, where $\mathrm{L}_e := C_\phi C_{\mathsf{Z}} \mathrm{L}_p + C_{\mathsf{S}} \mathrm{L}_\phi$.*

For *non-convex* optimization such as (1), it has been shown [Mairal, 2015, Proposition 3.1] that the incremental MM method converges asymptotically to a stationary solution of a problem. We strengthen their result by establishing a non-asymptotic rate, which is new to the literature.

**Theorem 1.** *Consider the iEM algorithm, i.e., Algorithm 1 with (7). Assume H1, H2, H3, H4. For any $K_{\max} \geq 1$, it holds that*

$$\mathbb{E}[\|\nabla \overline{\mathcal{L}}(\hat{\boldsymbol{\theta}}^{(K)})\|^2] \leq n \frac{2 \mathrm{L}_e}{K_{\max}} \mathbb{E}\big[\overline{\mathcal{L}}(\hat{\boldsymbol{\theta}}^{(0)}) - \overline{\mathcal{L}}(\hat{\boldsymbol{\theta}}^{(K_{\max})})\big], \tag{21}$$

*where $\mathrm{L}_e$ is defined in Lemma 1 and $K$ is a uniform random variable on $[\![0, K_{\max} - 1]\!]$ [cf. (11)] independent of the $\{i_k\}_{k=0}^{K_{\max}}$.*

We remark that under suitable assumptions, our analysis in Theorem 1 also extends to several non-exponential family distribution models.

## 3.2 Stochastic EM as Scaled Gradient Methods

We interpret the sEM-VR and fiEMmethods as *scaled gradient* methods on the sufficient statistics $\hat{\mathbf{s}}$, tackling a *non-convex* optimization problem. The benefit of doing so is that we are able to demonstrate a faster convergence rate for these methods through motivating them as *variance reduced* optimization methods. The latter is shown to be more effective when handling large datasets [Reddi et al., 2016b,a, Allen-Zhu and Hazan, 2016] than traditional stochastic/deterministic optimization methods. To set our stage, we consider the minimization problem:

$$\min_{\mathbf{s} \in \mathsf{S}} V(\mathbf{s}) := \overline{\mathcal{L}}(\overline{\boldsymbol{\theta}}(\mathbf{s})) = \mathrm{R}(\overline{\boldsymbol{\theta}}(\mathbf{s})) + \frac{1}{n} \sum_{i=1}^{n} \mathcal{L}_i(\overline{\boldsymbol{\theta}}(\mathbf{s})), \tag{22}$$

where $\overline{\boldsymbol{\theta}}(\mathbf{s})$ is the unique map defined in the M-step (5). We first show that the stationary points of (22) has a one-to-one correspondence with the stationary points of (1):

**Lemma 2.** *For any* $\mathbf{s} \in \mathsf{S}$*, it holds that*

$$\nabla_{\mathbf{s}} V(\mathbf{s}) = \mathrm{J}_{\overline{\boldsymbol{\theta}}}^{\mathbf{s}}(\mathbf{s})^{\top} \nabla_{\boldsymbol{\theta}} \overline{\mathcal{L}}(\overline{\boldsymbol{\theta}}(\mathbf{s})). \tag{23}$$

*Assume H4. If* $\mathbf{s}^{\star} \in \{\mathbf{s} \in \mathsf{S} : \nabla_{\mathbf{s}} V(\mathbf{s}) = 0\}$*, then* $\overline{\boldsymbol{\theta}}(\mathbf{s}^{\star}) \in \{\boldsymbol{\theta} \in \Theta : \nabla_{\boldsymbol{\theta}} \overline{\mathcal{L}}(\boldsymbol{\theta}) = 0\}$*. Conversely,
if* $\boldsymbol{\theta}^{*} \in \{\boldsymbol{\theta} \in \Theta : \nabla_{\boldsymbol{\theta}} \overline{\mathcal{L}}(\boldsymbol{\theta}) = 0\}$*, then* $\mathbf{s}^{*} = \overline{\mathbf{s}}(\boldsymbol{\theta}^{*}) \in \{\mathbf{s} \in \mathsf{S} : \nabla_{\mathbf{s}} V(\mathbf{s}) = 0\}$*.*

The next lemmas show that the update direction, $\hat{s}^{(k)} - \mathcal{S}^{(k+1)}$, in the sE-step (4) of sEM-VR and
fiEM methods is a *scaled gradient* of $V(\mathbf{s})$. We first observe the following conditional expectation:

$$\mathbb{E}\big[\hat{s}^{(k)} - \mathcal{S}^{(k+1)} | \mathcal{F}_k\big] = \hat{s}^{(k)} - \overline{\mathbf{s}}^{(k)} = \hat{s}^{(k)} - \overline{\mathbf{s}}(\overline{\boldsymbol{\theta}}(\hat{s}^{(k)})), \tag{24}$$

where $\mathcal{F}_k$ is the $\sigma$-algebra generated by $\{i_0, i_1, \ldots, i_k\}$ (or $\{i_0, j_0, \ldots, i_k, j_k\}$ for fiEM).

The difference vector $\mathbf{s} - \overline{\mathbf{s}}(\overline{\boldsymbol{\theta}}(\mathbf{s}))$ and the gradient vector $\nabla_{\mathbf{s}} V(\mathbf{s})$ are correlated, as we observe:

**Lemma 3.** *Assume H4,H5. For all* $\mathbf{s} \in \mathsf{S}$*,*

$$v_{\min}^{-1} \langle \nabla V(\mathbf{s}) \, | \, \mathbf{s} - \overline{\mathbf{s}}(\overline{\boldsymbol{\theta}}(\mathbf{s})) \rangle \geq \big\| \mathbf{s} - \overline{\mathbf{s}}(\overline{\boldsymbol{\theta}}(\mathbf{s})) \big\|^2 \geq v_{\max}^{-2} \| \nabla V(\mathbf{s}) \|^2, \tag{25}$$

Combined with (24), the above lemma shows that the update direction in the sE-step (4) of sEM-VR
and fiEM methods is a *stochastic scaled gradient* where $\hat{s}^{(k)}$ is updated with a stochastic direction
whose mean is correlated with $\nabla V(\mathbf{s})$.

Furthermore, the expectation step's operator and the objective function in (22) are smooth functions:

**Lemma 4.** *Assume H1, H3, H4, H5. For all* $\mathbf{s}, \mathbf{s}' \in \mathsf{S}$ *and* $i \in [\![1, n]\!]$*, we have*

$$\| \overline{\mathbf{s}}_i(\overline{\boldsymbol{\theta}}(\mathbf{s})) - \overline{\mathbf{s}}_i(\overline{\boldsymbol{\theta}}(\mathbf{s}')) \| \leq \mathrm{L}_{\mathbf{s}} \, \| \mathbf{s} - \mathbf{s}' \|, \quad \| \nabla V(\mathbf{s}) - \nabla V(\mathbf{s}') \| \leq \mathrm{L}_V \, \| \mathbf{s} - \mathbf{s}' \|, \tag{26}$$

*where* $\mathrm{L}_{\mathbf{s}} := C_{\mathsf{Z}} \, \mathrm{L}_p \, \mathrm{L}_\theta$ *and* $\mathrm{L}_V := v_{\max}(1 + \mathrm{L}_{\mathbf{s}}) + \mathrm{L}_B \, C_{\mathsf{S}}$.

The following theorem establishes the (fast) non-asymptotic convergence rates of sEM-VR and fiEM
methods, which are similar to [Reddi et al., 2016b,a, Allen-Zhu and Hazan, 2016]:

**Theorem 2.** *Assume H1, H3, H4, H5. Denote* $\overline{L}_{\mathsf{v}} = \max\{\mathrm{L}_V, \mathrm{L}_{\mathbf{s}}\}$ *with the constants in Lemma 4.*

- *Consider the sEM-VR method,* i.e., *Algorithm 1 with (9). There exists a universal constant* $\mu \in (0, 1)$ *(independent of* $n$*) such that if we set the step size as* $\gamma = \frac{\mu v_{\min}}{\overline{L}_{\mathsf{v}} n^{2/3}}$ *and the epoch length as* $m = \frac{n}{2\mu^2 v_{\min}^2 + \mu}$*, then for any* $K_{\mathsf{max}} \geq 1$ *that is a multiple of* $m$*, it holds that*

$$\mathbb{E}[\| \nabla V(\hat{s}^{(K)}) \|^2] \leq n^{\frac{2}{3}} \frac{2\overline{L}_{\mathsf{v}}}{\mu K_{\mathsf{max}}} \frac{v_{\max}^2}{v_{\min}^2} \mathbb{E}[V(\hat{s}^{(0)}) - V(\hat{s}^{(K_{\mathsf{max}})})]. \tag{27}$$

- *Consider the fiEM method,* i.e., *Algorithm 1 with (10). Set* $\gamma = \frac{v_{\min}}{\alpha \overline{L}_{\mathsf{v}} n^{2/3}}$ *such that* $\alpha = \max\{6, 1 + 4v_{\min}\}$*. For any* $K_{\mathsf{max}} \geq 1$*, it holds that*

$$\mathbb{E}[\| \nabla V(\hat{s}^{(K)}) \|^2] \leq n^{\frac{2}{3}} \frac{\alpha^2 \overline{L}_{\mathsf{v}}}{K_{\mathsf{max}}} \frac{v_{\max}^2}{v_{\min}^2} \mathbb{E}\big[V(\hat{s}^{(0)}) - V(\hat{s}^{(K_{\mathsf{max}})})\big]. \tag{28}$$

*We recall that* $K$ *in the above is a uniform and independent r.v. chosen from* $[\![K_{\mathsf{max}} - 1]\!]$ *[cf. (11)].*

In the supplementary materials, we also provide a local convergence analysis for fiEM method which
shows that the latter can achieve linear rate of convergence *locally* under a similar set of assumptions
used in [Chen et al., 2018] for sEM-VR method.

**Comparing iEM, sEM-VR, and fiEM**   Note that by (23) in Lemma 2, if $\| \nabla_{\mathbf{s}} V(\hat{\mathbf{s}}) \|^2 \leq \epsilon$, then
$\| \nabla_{\boldsymbol{\theta}} \overline{\mathcal{L}}(\overline{\boldsymbol{\theta}}(\hat{\mathbf{s}})) \|^2 = \mathcal{O}(\epsilon)$, and vice versa, where the hidden constant is independent of $n$. In other
words, the rates for iEM, sEM-VR, fiEM methods in Theorem 1 and 2 are comparable.

Importantly, the theorems show an intriguing comparison – to attain an $\epsilon$-stationary point with
$\| \nabla_{\boldsymbol{\theta}} \overline{\mathcal{L}}(\overline{\boldsymbol{\theta}}(\hat{\mathbf{s}})) \|^2 \leq \epsilon$ or $\| \nabla_{\mathbf{s}} V(\hat{\mathbf{s}}) \|^2 \leq \epsilon$, the iEM method requires $\mathcal{O}(n/\epsilon)$ iterations (in expec-
tation) while the sEM-VR, fiEM methods require only $\mathcal{O}(n^{\frac{2}{3}}/\epsilon)$ iterations (in expectation). This
comparison can be surprising since the iEM method is a monotone method as it guarantees decrease
in the objective value; while the sEM-VR, fiEM methods are non-monotone. Nevertheless, it aligns
with the recent analysis on stochastic variance reduction methods on non-convex problems. In the
next section, we confirm the theory by observing a similar behavior numerically.

# 4 Numerical Examples

## 4.1 Gaussian Mixture Models

As described in Section 2.1, our goal is to fit a GMM model to a set of $n$ observations $\{y_i\}_{i=1}^n$ whose distribution is modeled as a Gaussian mixture of $M$ components, each with a unit variance. Let $z_i \in [\![M]\!]$ be the latent labels, the complete log-likelihood is given in (12), where $\boldsymbol{\theta} := (\boldsymbol{\omega}, \boldsymbol{\mu})$ with $\boldsymbol{\omega} = \{\omega_m\}_{m=1}^{M-1}$ are the mixing weights with the convention $\omega_M = 1 - \sum_{m=1}^{M-1} \omega_m$ and $\boldsymbol{\mu} = \{\mu_m\}_{m=1}^M$ are the means. The constraint set on $\boldsymbol{\theta}$ is given by

$$\Theta = \{\omega_m, \ m = 1, ..., M-1 : \omega_m \geq 0, \ \textstyle\sum_{m=1}^{M-1} \omega_m \leq 1\} \times \{\mu_m \in \mathbb{R}, \ m = 1, ..., M\}. \quad (29)$$

In the following experiments of synthetic data, we generate samples from a GMM model with $M = 2$ components with two mixtures with means $\mu_1 = -\mu_2 = 0.5$, see Appendix G.1 for details of the implementation and satisfaction of model assumptions for GMM inference. We aim at verifying the theoretical results in Theorem 1 and 2 of the dependence on sample size $n$.

**Fixed sample size** We use $n = 10^4$ synthetic samples and run the bEM method until convergence (to double precision) to obtain the ML estimate $\mu^\star$. We compare the bEM, sEM, iEM, sEM-VR and fiEM methods in terms of their precision measured by $|\mu - \mu^\star|^2$. We set the stepsize of the sEM as $\gamma_k = 3/(k + 10)$, and the stepsizes of the sEM-VR and the fiEM to a constant stepsize proportional to $1/n^{2/3}$ and equal to $\gamma = 0.003$. The left plot of Figure 1 shows the convergence of the precision $|\mu - \mu^*|^2$ for the different methods against the epoch(s) elapsed (one epoch equals $n$ iterations). We observe that the sEM-VR and fiEM methods outperform the other methods, supporting our analytical results.

**Varying sample size** We compare the number of *iterations* required to reach a precision of $10^{-3}$ as a function of the sample size from $n = 10^3$ to $n = 10^5$. We average over 5 independent runs for each method using the same stepsizes as in the finite sample size case above. The right plot of Figure 1 confirms that our findings in Theorem 1 and 2 are sharp. It requires $\mathcal{O}(n/\epsilon)$ (*resp.* $\mathcal{O}(n^{\frac{2}{3}}/\epsilon)$) iterations to find a $\epsilon$-stationary point for the iEM (*resp.* sEM-VR and fiEM) method.

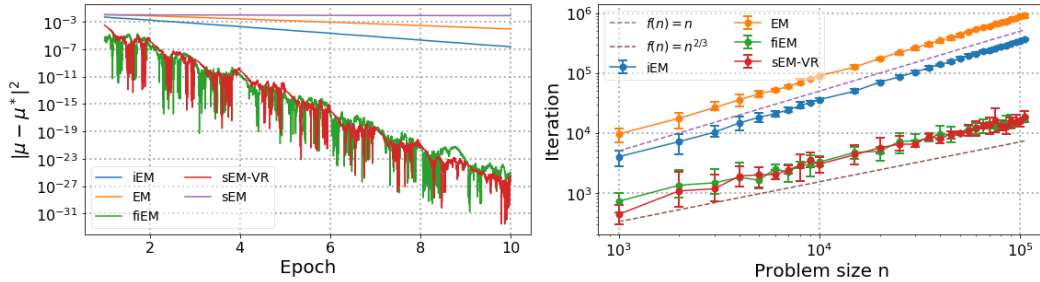

Figure 1: Performance of stochastic EM methods for fitting a GMM. (Left) Precision ($|\mu^{(k)} - \mu^\star|^2$) as a function of the epoch elapsed. (Right) Number of iterations to reach a precision of $10^{-3}$.

## 4.2 Probabilistic Latent Semantic Analysis

The second example considers probabilistic Latent Semantic Analysis (pLSA) whose aim is to classify documents into a number of topics. We are given a collection of documents $[\![D]\!]$ with terms from a vocabulary $[\![V]\!]$. The data is summarized by a list of tokens $\{y_i\}_{i=1}^n$ where each token is a pair of document and word $y_i = (y_i^{(\mathrm{d})}, y_i^{(\mathrm{w})})$ which indicates that $y_i^{(\mathrm{w})}$ appears in document $y_i^{(\mathrm{d})}$. The goal of pLSA is to classify the documents into $K$ topics, which is modeled as a latent variable $z_i \in [\![K]\!]$ associated with each token [Hofmann, 1999].

To apply stochastic EM methods for pLSA, we define $\boldsymbol{\theta} := (\boldsymbol{\theta}^{(\mathrm{t|d})}, \boldsymbol{\theta}^{(\mathrm{w|t})})$ as the parameter variable, where $\boldsymbol{\theta}^{(\mathrm{t|d})} = \{\boldsymbol{\theta}_{d,k}^{(\mathrm{t|d})}\}_{[\![K-1]\!] \times [\![D]\!]}$ and $\boldsymbol{\theta}^{(\mathrm{w|t})} = \{\boldsymbol{\theta}_{k,v}^{(\mathrm{w|t})}\}_{[\![K]\!] \times [\![V-1]\!]}$. The constraint set $\Theta$ is given as — for each $d \in [\![D]\!]$, $\boldsymbol{\theta}_{d,\cdot}^{(\mathrm{t|d})} \in \Delta^K$ and for each $k \in [\![K]\!]$, we have $\boldsymbol{\theta}_{\cdot,k}^{(\mathrm{w|t})} \in \Delta^V$, where $\Delta^K$, $\Delta^V$

are the (reduced dimension) $K, V$-dimensional probability simplex; see (108) in the supplementary material for the precise definition. Furthermore, denote $\boldsymbol{\theta}_{d,K}^{(\mathrm{t}|\mathrm{d})} = 1 - \sum_{k=1}^{K-1} \boldsymbol{\theta}_{d,k}^{(\mathrm{t}|\mathrm{d})}$ for each $d \in [\![D]\!]$, and $\boldsymbol{\theta}_{k,V}^{(\mathrm{w}|\mathrm{t})} = 1 - \sum_{\ell=1}^{V-1} \boldsymbol{\theta}_{k,\ell}^{(\mathrm{w}|\mathrm{t})}$ for each $k \in [\![K]\!]$, the complete log likelihood for $(y_i, z_i)$ is (up to an additive constant term):

$$\log f(z_i, y_i; \boldsymbol{\theta}) = \sum_{k=1}^{K} \sum_{d=1}^{D} \log(\boldsymbol{\theta}_{d,k}^{(\mathrm{t}|\mathrm{d})}) \mathbb{1}_{\{k,d\}}(z_i, y_i^{(\mathrm{d})}) + \sum_{k=1}^{K} \sum_{v=1}^{V} \log(\boldsymbol{\theta}_{k,v}^{(\mathrm{w}|\mathrm{t})}) \mathbb{1}_{\{k,v\}}(z_i, y_i^{(\mathrm{w})}). \quad (30)$$

The penalization function is designed as

$$\mathrm{R}(\boldsymbol{\theta}^{(\mathrm{t}|\mathrm{d})}, \boldsymbol{\theta}^{(\mathrm{w}|\mathrm{t})}) = -\log \mathrm{Dir}(\boldsymbol{\theta}^{(\mathrm{t}|\mathrm{d})}; K, \alpha') - \log \mathrm{Dir}(\boldsymbol{\theta}^{(\mathrm{w}|\mathrm{t})}; V, \beta'), \quad (31)$$

such that we ensure $\overline{\boldsymbol{\theta}}(\boldsymbol{s}) \in \mathrm{int}(\Theta)$. We can apply the stochastic EM methods described in Section 2 on the pLSA problem. The implementation details are provided in Appendix G.2, therein we also verify the model assumptions required by our convergence analysis for pLSA.

**Experiment** We compare the stochastic EM methods on two FAO (UN Food and Agriculture Organization) datasets [Medelyan, 2009]. The first (*resp.* second) dataset consists of $10^3$ (*resp.* $10.5 \times 10^3$) documents and a vocabulary of size 300. The number of topics is set to $K = 10$ and the stepsizes for the fiEM and sEM-VR are set to $\gamma = 1/n^{2/3}$ while the stepsize for the sEM is set to $\gamma_k = 1/(k+10)$. Figure 1 shows the evidence lower bound (ELBO) as a function of the number of epochs for the datasets. Again, the result shows that fiEM and sEM-VR methods achieve faster convergence than the competing EM methods, affirming our theoretical findings.

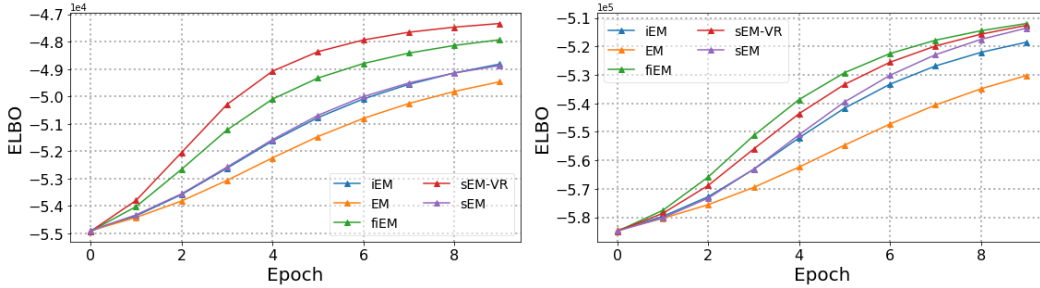

Figure 2: ELBO of the stochastic EM methods on FAO datasets as a function of number of epochs elapsed. (Left) small dataset with $10^3$ documents. (Right) large dataset with $10.5 \times 10^3$ documents.

## 5  Conclusion

This paper studies the global convergence for stochastic EM methods. Particularly, we focus on the inference of latent variable model with exponential family distribution and analyze the convergence of several stochastic EM methods. Our convergence results are *global* and *non-asymptotic*, and we offer two complimentary views on the existing stochastic EM methods — one interprets iEM method as an incremental MM method, and one interprets sEM-VR and fiEM methods as scaled gradient methods. The analysis shows that the sEM-VR and fiEM methods converge faster than the iEM method, and the result is confirmed via numerical experiments.

## Acknowledgement

BK and HTW contributed equally to this work. HTW's work is supported by the CUHK Direct Grant #4055113.

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
