[Supplementary Material]

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

# A Proof of Lemma 1

**Lemma.** *Assume H1, H2, H3, H4. Let $e_i(\boldsymbol{\theta}; \boldsymbol{\theta}') := Q_i(\boldsymbol{\theta}; \boldsymbol{\theta}') - \mathcal{L}_i(\boldsymbol{\theta})$. For any $\boldsymbol{\theta}, \bar{\boldsymbol{\theta}}, \boldsymbol{\theta}' \in \Theta^3$, we have $\|\nabla e_i(\boldsymbol{\theta}; \boldsymbol{\theta}') - \nabla e_i(\bar{\boldsymbol{\theta}}; \boldsymbol{\theta}')\| \leq \mathrm{L}_e \|\boldsymbol{\theta} - \bar{\boldsymbol{\theta}}\|$, where $\mathrm{L}_e := C_\phi C_{\mathsf{Z}} \mathrm{L}_p + C_{\mathsf{S}} \mathrm{L}_\phi$.*

**Proof** Observe the following identity

$$
\begin{aligned}
\nabla_{\boldsymbol{\theta}} \mathcal{L}_i(\boldsymbol{\theta})\big|_{\boldsymbol{\theta}=\hat{\boldsymbol{\theta}}} &= \nabla_{\boldsymbol{\theta}} \Big\{ -\log \int_{\mathsf{Z}} f(z_i, y_i; \boldsymbol{\theta}) \mu(\mathrm{d}z_i) \Big\}\Big|_{\boldsymbol{\theta}=\hat{\boldsymbol{\theta}}} \\
&\overset{(a)}{=} -\int_{\mathsf{Z}} \{\nabla_{\boldsymbol{\theta}} \log f(z_i, y_i; \boldsymbol{\theta})\}\big|_{\boldsymbol{\theta}=\hat{\boldsymbol{\theta}}} f(z_i|y_i; \hat{\boldsymbol{\theta}}) \mu(\mathrm{d}z_i) \\
&= \nabla_{\boldsymbol{\theta}} Q_i(\boldsymbol{\theta}; \hat{\boldsymbol{\theta}})\big|_{\boldsymbol{\theta}=\hat{\boldsymbol{\theta}}}
\end{aligned}
\tag{32}
$$

where (a) is due to the Fisher's identity and (b) is due to the definition of $Q_i$ in (18). It follows that

$$
\nabla e_i(\boldsymbol{\theta}; \hat{\boldsymbol{\theta}}) = \nabla \{Q_i(\boldsymbol{\theta}, \hat{\boldsymbol{\theta}}) - \mathcal{L}_i(\boldsymbol{\theta})\} = \mathrm{J}_\phi^{\boldsymbol{\theta}}(\boldsymbol{\theta})^\top (\bar{\mathbf{s}}_i(\hat{\boldsymbol{\theta}}) - \bar{\mathbf{s}}_i(\boldsymbol{\theta})).
\tag{33}
$$

We observe that

$$
\begin{aligned}
\|\bar{\mathbf{s}}_i(\boldsymbol{\theta}) - \bar{\mathbf{s}}_i(\boldsymbol{\theta}')\| &= \left\| \int_{\mathsf{Z}} S(z_i, y_i) \{p(z_i|y_i; \boldsymbol{\theta}) - p(z_i|y_i; \boldsymbol{\theta}')\} \mu(\mathrm{d}z_i) \right\| \\
&\leq \mathrm{L}_p \|\boldsymbol{\theta} - \boldsymbol{\theta}'\| \int_{\mathsf{Z}} |S(z_i, y_i)| \mu(\mathrm{d}z_i) \leq C_{\mathsf{Z}} \mathrm{L}_p \|\boldsymbol{\theta} - \boldsymbol{\theta}'\|
\end{aligned}
\tag{34}
$$

where the last inequality is due to the compactness of $\mathsf{Z}$. Finally, we have

$$
\begin{aligned}
\|\nabla e_i(\boldsymbol{\theta}; \hat{\boldsymbol{\theta}}) - \nabla e_i(\bar{\boldsymbol{\theta}}; \hat{\boldsymbol{\theta}})\| &\leq \| \mathrm{J}_\phi^{\boldsymbol{\theta}}(\boldsymbol{\theta})\| \|\bar{\mathbf{s}}_i(\boldsymbol{\theta}) - \bar{\mathbf{s}}_i(\bar{\boldsymbol{\theta}})\| + \| \mathrm{J}_\phi^{\boldsymbol{\theta}}(\boldsymbol{\theta}) - \mathrm{J}_\phi^{\boldsymbol{\theta}}(\bar{\boldsymbol{\theta}})\| \|\bar{\mathbf{s}}_i(\hat{\boldsymbol{\theta}}) - \bar{\mathbf{s}}_i(\bar{\boldsymbol{\theta}})\| \\
&\leq (C_\phi C_{\mathsf{Z}} \mathrm{L}_p + C_{\mathsf{S}} \mathrm{L}_\phi) \|\boldsymbol{\theta} - \bar{\boldsymbol{\theta}}\|
\end{aligned}
\tag{35}
$$

where the last inequality is due to the compactness of $\mathsf{S}$. $\qquad\square$

# B Proof of Theorem 1

**Theorem.** *Consider the iEM algorithm,* i.e., *Algorithm 1 with (7). Assume H1, H2, H3, H4. For any $K_{\max} \geq 1$, it holds that*

$$
\mathbb{E}[\|\nabla \overline{\mathcal{L}}(\hat{\boldsymbol{\theta}}^{(K)})\|^2] \leq n \frac{2 \mathrm{L}_e}{K_{\max}} \mathbb{E}\big[\overline{\mathcal{L}}(\hat{\boldsymbol{\theta}}^{(0)}) - \overline{\mathcal{L}}(\hat{\boldsymbol{\theta}}^{(K_{\max})})\big],
$$

*where $\mathrm{L}_e$ is defined in Lemma 1 and $K$ is a uniform random variable on $[\![0, K_{\max} - 1]\!]$ [cf. (11)] independent of the $\{i_k\}_{k=0}^{K_{\max}}$.*

**Proof** We derive a *non-asymptotic* convergence rate for the iEM method. To begin our analysis, define

$$
\overline{\mathcal{L}}^{(k+1)}(\boldsymbol{\theta}) := \mathrm{R}(\boldsymbol{\theta}) + \frac{1}{n} \sum_{i=1}^n Q_i(\boldsymbol{\theta}; \hat{\boldsymbol{\theta}}^{(\tau_i^{k+1})})
\tag{36}
$$

One has

$$
\overline{\mathcal{L}}^{(k+1)}(\boldsymbol{\theta}) = \overline{\mathcal{L}}^{(k)}(\boldsymbol{\theta}) + \frac{1}{n}\big(Q_{i_k}(\boldsymbol{\theta}; \hat{\boldsymbol{\theta}}^{(k)}) - Q_{i_k}(\boldsymbol{\theta}; \hat{\boldsymbol{\theta}}^{(\tau_{i_k}^k)})\big)
\tag{37}
$$

Observe that $\hat{\boldsymbol{\theta}}^{(k+1)} \in \arg\min_{\boldsymbol{\theta} \in \Theta} \overline{\mathcal{L}}^{(k+1)}(\boldsymbol{\theta})$. We have

$$
\begin{aligned}
\overline{\mathcal{L}}^{(k+1)}(\hat{\boldsymbol{\theta}}^{(k+1)}) \leq \overline{\mathcal{L}}^{(k+1)}(\hat{\boldsymbol{\theta}}^{(k)}) &= \overline{\mathcal{L}}^{(k)}(\hat{\boldsymbol{\theta}}^{(k)}) + \frac{1}{n}\big(Q_{i_k}(\hat{\boldsymbol{\theta}}^{(k)}; \hat{\boldsymbol{\theta}}^{(k)}) - Q_{i_k}(\hat{\boldsymbol{\theta}}^{(k)}; \hat{\boldsymbol{\theta}}^{(\tau_{i_k}^k)})\big) \\
&= \overline{\mathcal{L}}^{(k)}(\hat{\boldsymbol{\theta}}^{(k)}) + \frac{1}{n}\big(\mathcal{L}_{i_k}(\hat{\boldsymbol{\theta}}^{(k)}) - Q_{i_k}(\hat{\boldsymbol{\theta}}^{(k)}; \hat{\boldsymbol{\theta}}^{(\tau_{i_k}^k)})\big)
\end{aligned}
\tag{38}
$$

where we have used the identity $\mathcal{L}_{i_k}(\hat{\boldsymbol{\theta}}^{(k)}) = Q_{i_k}(\hat{\boldsymbol{\theta}}^{(k)}; \hat{\boldsymbol{\theta}}^{(k)})$. Arranging terms imply

$$
e_{i_k}(\hat{\boldsymbol{\theta}}^{(k)}; \hat{\boldsymbol{\theta}}^{(\tau_{i_k}^k)}) = Q_{i_k}(\hat{\boldsymbol{\theta}}^{(k)}; \hat{\boldsymbol{\theta}}^{(\tau_{i_k}^k)}) - \mathcal{L}_{i_k}(\hat{\boldsymbol{\theta}}^{(k)}) \leq n\big(\overline{\mathcal{L}}^{(k)}(\hat{\boldsymbol{\theta}}^{(k)}) - \overline{\mathcal{L}}^{(k+1)}(\hat{\boldsymbol{\theta}}^{(k+1)})\big)
\tag{39}
$$

For $k \in \mathbb{N}^*$, denote by $\mathcal{F}_k$ the $\sigma$-algebra generated by the random variables $i_0, \ldots, i_{k-1}$. Note that $\hat{\boldsymbol{\theta}}^{(k)}$ is $\mathcal{F}_k$-measurable. Because the random variable $i_k$ is independent of $\mathcal{F}_{k-1}$ and is uniformly distributed over $\{1, \ldots, n\}$, the conditional expectation evaluates to

$$\mathbb{E}\left[ e_{i_k}(\hat{\boldsymbol{\theta}}^{(k)}; \hat{\boldsymbol{\theta}}^{(\tau_{i_k}^k)}) \,\Big|\, \mathcal{F}_k \right] = \overline{\mathcal{L}}^{(k)}(\hat{\boldsymbol{\theta}}^{(k)}) - \overline{\mathcal{L}}(\hat{\boldsymbol{\theta}}^{(k)}) \tag{40}$$

where $\overline{\mathcal{L}}$ is the global objective function defined in (1). Note that the function $\overline{\mathcal{L}}^{(k)}(\boldsymbol{\theta}) - \overline{\mathcal{L}}(\boldsymbol{\theta})$ is non-negative and $\mathrm{L}_e$-smooth. It follows that for any $\boldsymbol{\theta}$, the inequality holds

$$0 \le \overline{\mathcal{L}}^{(k)}(\boldsymbol{\theta}) - \overline{\mathcal{L}}(\boldsymbol{\theta}) \le \overline{\mathcal{L}}^{(k)}(\hat{\boldsymbol{\theta}}^{(k)}) - \overline{\mathcal{L}}(\hat{\boldsymbol{\theta}}^{(k)}) - \langle \nabla\overline{\mathcal{L}}(\hat{\boldsymbol{\theta}}^{(k)}) \,|\, \boldsymbol{\theta} - \hat{\boldsymbol{\theta}}^{(k)} \rangle + \frac{\mathrm{L}_e}{2}\|\boldsymbol{\theta} - \hat{\boldsymbol{\theta}}^{(k)}\|^2, \tag{41}$$

where we have used the fact $\nabla\overline{\mathcal{L}}^{(k)}(\hat{\boldsymbol{\theta}}^{(k)}) = \mathbf{0}$. Setting $\boldsymbol{\theta} = \hat{\boldsymbol{\theta}}^{(k)} + (\mathrm{L}_e)^{-1}\nabla\overline{\mathcal{L}}(\hat{\boldsymbol{\theta}}^{(k)})$ in the above yields

$$\frac{1}{2\,\mathrm{L}_e}\|\nabla\overline{\mathcal{L}}(\hat{\boldsymbol{\theta}}^{(k)})\|^2 \le \overline{\mathcal{L}}^{(k)}(\hat{\boldsymbol{\theta}}^{(k)}) - \overline{\mathcal{L}}(\hat{\boldsymbol{\theta}}^{(k)}) \tag{42}$$

Therefore, taking the conditional expectation on both sides of (39) leads to

$$\frac{1}{2n\,\mathrm{L}_e}\|\nabla\overline{\mathcal{L}}(\hat{\boldsymbol{\theta}}^{(k)})\|^2 \le \overline{\mathcal{L}}^{(k)}(\hat{\boldsymbol{\theta}}^{(k)}) - \mathbb{E}\big[\overline{\mathcal{L}}^{(k+1)}(\hat{\boldsymbol{\theta}}^{(k+1)})|\mathcal{F}_k\big] \tag{43}$$

Note that as we have set $\gamma_{k+1} = 1$ in the iEM method, the terminating iteration number $K$ is chosen uniformly over $\{1, \ldots, K_{\mathsf{max}}\}$, therefore taking the total expectations gives

$$\begin{aligned}
\mathbb{E}\big[\|\nabla\overline{\mathcal{L}}(\hat{\boldsymbol{\theta}}^{(K)})\|^2\big] &= \frac{1}{K_{\mathsf{max}}} \sum_{k=0}^{K_{\mathsf{max}}-1} \mathbb{E}\big[\|\nabla\overline{\mathcal{L}}(\hat{\boldsymbol{\theta}}^{(k)})\|^2\big] \\
&\le \frac{2n\,\mathrm{L}_e}{K_{\mathsf{max}}}\mathbb{E}\Big[\overline{\mathcal{L}}^{(0)}(\hat{\boldsymbol{\theta}}^{(0)}) - \overline{\mathcal{L}}^{(K_{\mathsf{max}})}(\hat{\boldsymbol{\theta}}^{(K_{\mathsf{max}}+1)})\Big] \\
&\le \frac{2n\,\mathrm{L}_e}{K_{\mathsf{max}}}\mathbb{E}\Big[\overline{\mathcal{L}}^{(0)}(\hat{\boldsymbol{\theta}}^{(0)}) - \overline{\mathcal{L}}(\hat{\boldsymbol{\theta}}^{(K_{\mathsf{max}})})\Big]
\end{aligned} \tag{44}$$

Lastly, we note that $\overline{\mathcal{L}}(\hat{\boldsymbol{\theta}}^{(0)}) = \overline{\mathcal{L}}^{(0)}(\hat{\boldsymbol{\theta}}^{(0)})$. This leads to (21) and concludes our proof. $\qquad\square$

## C   Proof of Lemma 2

**Lemma.** *For any* $\mathbf{s} \in \mathsf{S}$*, it holds that*

$$\nabla_{\mathbf{s}} V(\mathbf{s}) = \mathrm{J}_{\overline{\boldsymbol{\theta}}}^{\mathbf{s}}(\mathbf{s})^\top \nabla_{\boldsymbol{\theta}} \overline{\mathcal{L}}(\overline{\boldsymbol{\theta}}(\mathbf{s})).$$

*Assume H4. If* $\mathbf{s}^\star \in \{\mathbf{s} \in \mathsf{S} : \nabla_{\mathbf{s}} V(\mathbf{s}) = 0\}$*, then* $\overline{\boldsymbol{\theta}}(\mathbf{s}^\star) \in \{\boldsymbol{\theta} \in \Theta : \nabla_{\boldsymbol{\theta}}\overline{\mathcal{L}}(\boldsymbol{\theta}) = 0\}$*. Conversely, if* $\boldsymbol{\theta}^* \in \{\boldsymbol{\theta} \in \Theta : \nabla_{\boldsymbol{\theta}}\overline{\mathcal{L}}(\boldsymbol{\theta}) = 0\}$*, then* $\mathbf{s}^* = \overline{\mathbf{s}}(\boldsymbol{\theta}^*) \in \{\mathbf{s} \in \mathsf{S} : \nabla_{\mathbf{s}} V(\mathbf{s}) = 0\}$*.*

**Proof** Using chain rule, we obtain $\nabla_{\mathbf{s}} V(\mathbf{s}) = \mathrm{J}_{\overline{\boldsymbol{\theta}}}^{\mathbf{s}}(\mathbf{s})^\top \nabla_{\boldsymbol{\theta}}\overline{\mathcal{L}}(\overline{\boldsymbol{\theta}}(\mathbf{s}))$ Obviously if $\nabla_{\mathbf{s}} V(\mathbf{s}^\star) = 0$, then $\nabla_{\boldsymbol{\theta}}\overline{\mathcal{L}}(\overline{\boldsymbol{\theta}}(\mathbf{s}^\star)) = 0$ because $\mathrm{J}_{\overline{\boldsymbol{\theta}}}^{\mathbf{s}}(\mathbf{s})$ is invertible. Consider now the converse. By the Fisher identity, we get $\nabla_{\boldsymbol{\theta}}\mathcal{L}_i(\boldsymbol{\theta}) = \nabla_{\boldsymbol{\theta}}\psi(\boldsymbol{\theta}) - \mathrm{J}_{\phi}^{\boldsymbol{\theta}}(\boldsymbol{\theta})^\top \overline{\mathbf{s}}_i(\boldsymbol{\theta})$ which implies that $\nabla_{\boldsymbol{\theta}}\overline{\mathcal{L}}(\boldsymbol{\theta}) = \nabla_{\boldsymbol{\theta}}\mathrm{R}(\boldsymbol{\theta}) + \nabla_{\boldsymbol{\theta}}\psi(\boldsymbol{\theta}) - \mathrm{J}_{\phi}^{\boldsymbol{\theta}}(\boldsymbol{\theta})^\top \overline{\mathbf{s}}(\boldsymbol{\theta})$. Hence, if $\nabla_{\boldsymbol{\theta}}\overline{\mathcal{L}}(\boldsymbol{\theta}^*) = 0$, then $\nabla_{\boldsymbol{\theta}}\mathrm{R}(\boldsymbol{\theta}^*) + \nabla_{\boldsymbol{\theta}}\psi(\boldsymbol{\theta}^*) - \mathrm{J}_{\phi}^{\boldsymbol{\theta}}(\boldsymbol{\theta}^*)^\top \overline{\mathbf{s}}^* = 0$ where we have set $\overline{\mathbf{s}}^* = \overline{\mathbf{s}}(\boldsymbol{\theta}^*)$. Under H4, the latter relation implies that $\boldsymbol{\theta}^* = \overline{\boldsymbol{\theta}}(\mathbf{s}^*)$. The proof follows. $\quad\square$

## D   Proof of Lemma 3

**Lemma.** *Assume H4,H5. For all* $\mathbf{s} \in \mathsf{S}$*,*

$$\upsilon_{\min}^{-1}\big\langle \nabla V(\mathbf{s}) \,|\, \mathbf{s} - \overline{\mathbf{s}}(\overline{\boldsymbol{\theta}}(\mathbf{s})) \big\rangle \ge \big\|\mathbf{s} - \overline{\mathbf{s}}(\overline{\boldsymbol{\theta}}(\mathbf{s}))\big\|^2 \ge \upsilon_{\max}^{-2}\|\nabla V(\mathbf{s})\|^2,$$

**Proof** Using H4 and the fact that we can exchange integration with differentiation and the Fisher's identity, we obtain

$$
\begin{aligned}
\nabla_{\mathbf{s}} V(\mathbf{s}) &= \mathrm{J}_{\overline{\boldsymbol{\theta}}}^{\mathbf{s}}(\mathbf{s})^{\top} \left( \nabla_{\boldsymbol{\theta}} \mathrm{R}(\overline{\boldsymbol{\theta}}(\mathbf{s})) + \nabla_{\boldsymbol{\theta}} \mathcal{L}(\overline{\boldsymbol{\theta}}(\mathbf{s})) \right) \\
&= \mathrm{J}_{\overline{\boldsymbol{\theta}}}^{\mathbf{s}}(\mathbf{s})^{\top} \left( \nabla_{\boldsymbol{\theta}} \psi(\overline{\boldsymbol{\theta}}(\mathbf{s})) + \nabla_{\boldsymbol{\theta}} \mathrm{R}(\overline{\boldsymbol{\theta}}(\mathbf{s})) - \mathrm{J}_{\phi}^{\boldsymbol{\theta}}(\overline{\boldsymbol{\theta}}(\mathbf{s}))^{\top} \overline{\mathbf{s}}(\overline{\boldsymbol{\theta}}(\mathbf{s})) \right) \\
&= \mathrm{J}_{\overline{\boldsymbol{\theta}}}^{\mathbf{s}}(\mathbf{s})^{\top} \mathrm{J}_{\phi}^{\boldsymbol{\theta}}(\overline{\boldsymbol{\theta}}(\mathbf{s}))^{\top} (\mathbf{s} - \overline{\mathbf{s}}(\overline{\boldsymbol{\theta}}(\mathbf{s}))) ,
\end{aligned}
\tag{45}
$$

Consider the following vector map:

$$
\mathbf{s} \to \nabla_{\boldsymbol{\theta}} L(\mathbf{s}, \boldsymbol{\theta})|_{\boldsymbol{\theta}=\overline{\boldsymbol{\theta}}(\mathbf{s})} = \nabla_{\boldsymbol{\theta}} \psi(\overline{\boldsymbol{\theta}}(\mathbf{s})) + \nabla_{\boldsymbol{\theta}} \mathrm{R}(\overline{\boldsymbol{\theta}}(\mathbf{s})) - \mathrm{J}_{\phi}^{\boldsymbol{\theta}}(\overline{\boldsymbol{\theta}}(\mathbf{s}))^{\top} \mathbf{s} .
\tag{46}
$$

Taking the gradient of the above map *w.r.t.* $\mathbf{s}$ and using assumption H4, we show that:

$$
\mathbf{0} = - \mathrm{J}_{\phi}^{\boldsymbol{\theta}}(\overline{\boldsymbol{\theta}}(\mathbf{s})) + \Big( \underbrace{\nabla_{\boldsymbol{\theta}}^{2} \big( \psi(\boldsymbol{\theta}) + \mathrm{R}(\boldsymbol{\theta}) - \langle \phi(\boldsymbol{\theta}) \,|\, \mathbf{s} \rangle \big)|_{\boldsymbol{\theta}=\overline{\boldsymbol{\theta}}(\mathbf{s})}}_{=\mathrm{H}_{L}^{\boldsymbol{\theta}}(\mathbf{s}; \boldsymbol{\theta})} \Big) \mathrm{J}_{\overline{\boldsymbol{\theta}}}^{\mathbf{s}}(\mathbf{s}) .
\tag{47}
$$

The above yields

$$
\nabla_{\mathbf{s}} V(\mathbf{s}) = \mathrm{B}(\mathbf{s})(\mathbf{s} - \overline{\mathbf{s}}(\overline{\boldsymbol{\theta}}(\mathbf{s})))
\tag{48}
$$

where we recall $\mathrm{B}(\mathbf{s}) = \mathrm{J}_{\phi}^{\boldsymbol{\theta}}(\overline{\boldsymbol{\theta}}(\mathbf{s})) \big( \mathrm{H}_{L}^{\boldsymbol{\theta}}(\mathbf{s}; \overline{\boldsymbol{\theta}}(\mathbf{s})) \big)^{-1} \mathrm{J}_{\phi}^{\boldsymbol{\theta}}(\overline{\boldsymbol{\theta}}(\mathbf{s}))^{\top}$. The proof of (25) follows directly from the assumption H5. $\square$

# E   Proof of Lemma 4

**Lemma.** *Assume H1, H3, H4, H5. For all $\mathbf{s}, \mathbf{s}' \in \mathsf{S}$ and $i \in [\![1, n]\!]$, we have*

$$
\|\overline{\mathbf{s}}_i(\overline{\boldsymbol{\theta}}(\mathbf{s})) - \overline{\mathbf{s}}_i(\overline{\boldsymbol{\theta}}(\mathbf{s}'))\| \leq \mathrm{L}_{\mathbf{s}} \|\mathbf{s} - \mathbf{s}'\|, \quad \|\nabla V(\mathbf{s}) - \nabla V(\mathbf{s}')\| \leq \mathrm{L}_V \|\mathbf{s} - \mathbf{s}'\|,
$$

*where $\mathrm{L}_{\mathbf{s}} := C_{\mathsf{Z}} \mathrm{L}_p \mathrm{L}_{\theta}$ and $\mathrm{L}_V := \upsilon_{\max}(1 + \mathrm{L}_{\mathbf{s}}) + \mathrm{L}_B C_{\mathsf{S}}$.*

**Proof** We prove the first inequality of the lemma in (26). Observe that

$$
\overline{\mathbf{s}}_i(\overline{\boldsymbol{\theta}}(\mathbf{s})) - \overline{\mathbf{s}}_i(\overline{\boldsymbol{\theta}}(\mathbf{s}')) = \int_{\mathsf{Z}} S(z_i, y_i) \big\{ p(z_i | y_i; \overline{\boldsymbol{\theta}}(\mathbf{s})) - p(z_i | y_i; \overline{\boldsymbol{\theta}}(\mathbf{s}')) \big\} \mu(\mathrm{d}z_i)
\tag{49}
$$

Taking norms on both sides and using H1, H3 yield

$$
\|\overline{\mathbf{s}}_i(\overline{\boldsymbol{\theta}}(\mathbf{s})) - \overline{\mathbf{s}}_i(\overline{\boldsymbol{\theta}}(\mathbf{s}'))\| \leq \mathrm{L}_p \|\overline{\boldsymbol{\theta}}(\mathbf{s}) - \overline{\boldsymbol{\theta}}(\mathbf{s}')\| \int_{\mathsf{Z}} |S(z_i, y_i)| \mu(\mathrm{d}z_i) \leq C_{\mathsf{Z}} \mathrm{L}_p \|\overline{\boldsymbol{\theta}}(\mathbf{s}) - \overline{\boldsymbol{\theta}}(\mathbf{s}')\|,
\tag{50}
$$

where we have $\int_{\mathsf{Z}} |S(z_i, y_i)| \mu(\mathrm{d}z_i) \leq C_{\mathsf{Z}}$. Furthermore, under H4, as $\overline{\boldsymbol{\theta}}(\mathbf{s})$ is Lipschitz, there exists $\mathrm{L}_{\theta}$ such that

$$
\|\overline{\boldsymbol{\theta}}(\mathbf{s}) - \overline{\boldsymbol{\theta}}(\mathbf{s}')\| \leq \mathrm{L}_{\theta} \|\mathbf{s} - \mathbf{s}'\|
\tag{51}
$$

Substituting back into (50) concludes the proof with $\mathrm{L}_{\mathbf{s}} = C_{\mathsf{Z}} \mathrm{L}_p \mathrm{L}_{\theta}$.

To prove the second inequality in (26), we observe that:

$$
\nabla_{\mathbf{s}} V(\mathbf{s}) = \mathrm{B}(\mathbf{s})(\mathbf{s} - \overline{\mathbf{s}}(\overline{\boldsymbol{\theta}}(\mathbf{s})))
\tag{52}
$$

We observe the upper bound

$$
\begin{aligned}
&\|\nabla V(\mathbf{s}) - \nabla V(\mathbf{s}')\| \\
&= \| \mathrm{B}(\mathbf{s})((\mathbf{s} - \overline{\mathbf{s}}(\overline{\boldsymbol{\theta}}(\mathbf{s}))) - (\mathbf{s}' - \overline{\mathbf{s}}(\overline{\boldsymbol{\theta}}(\mathbf{s}')))) + (\mathrm{B}(\mathbf{s}) - \mathrm{B}(\mathbf{s}'))(\mathbf{s}' - \overline{\mathbf{s}}(\overline{\boldsymbol{\theta}}(\mathbf{s}')))\| \\
&\leq \| \mathrm{B}(\mathbf{s})\| \|\mathbf{s} - \overline{\mathbf{s}}(\overline{\boldsymbol{\theta}}(\mathbf{s})) - (\mathbf{s}' - \overline{\mathbf{s}}(\overline{\boldsymbol{\theta}}(\mathbf{s}')))\| + \| \mathrm{B}(\mathbf{s}) - \mathrm{B}(\mathbf{s}')\| \|\mathbf{s}' - \overline{\mathbf{s}}(\overline{\boldsymbol{\theta}}(\mathbf{s}'))\|
\end{aligned}
\tag{53}
$$

We observe that

$$
\|\overline{\mathbf{s}}(\overline{\boldsymbol{\theta}}(\mathbf{s})) - \overline{\mathbf{s}}(\overline{\boldsymbol{\theta}}(\mathbf{s}'))\| \leq \frac{1}{n} \sum_{i=1}^{n} \|\overline{\mathbf{s}}_i(\overline{\boldsymbol{\theta}}(\mathbf{s})) - \overline{\mathbf{s}}_i(\overline{\boldsymbol{\theta}}(\mathbf{s}'))\| \leq \mathrm{L}_{\mathbf{s}} \|\mathbf{s} - \mathbf{s}'\|,
\tag{54}
$$

which is due to (26). Furthermore, as $\mathbf{s}' \in \mathsf{S}$, a compact set, we have $\|\mathbf{s}' - \overline{\mathbf{s}}(\overline{\boldsymbol{\theta}}(\mathbf{s}'))\| \leq C_{\mathsf{S}}$. Consequently, using H5 we have

$$
\|\nabla V(\mathbf{s}) - \nabla V(\mathbf{s}')\| \leq \Big( \upsilon_{\max}(1 + \mathrm{L}_{\mathbf{s}}) + \mathrm{L}_B C_{\mathsf{S}} \Big) \|\mathbf{s} - \mathbf{s}'\|,
\tag{55}
$$

which proves our claim. $\square$

# F  Proof of Theorem 2

**Theorem.** *Assume H1, H3, H4, H5. Denote $\overline{L}_{\mathsf{v}} = \max\{L_V, L_{\mathbf{s}}\}$ with the constants in Lemma 4.*

- *Consider the sEM-VR method, i.e., Algorithm 1 with (9). There exists a universal constant $\mu \in (0, 1)$ (independent of $n$) such that if we set the step size as $\gamma = \frac{\mu v_{\min}}{\overline{L}_{\mathsf{v}} n^{2/3}}$ and the epoch length as $m = \frac{n}{2\mu^2 v_{\min}^2 + \mu}$, then for any $K_{\mathsf{max}} \geq 1$ that is a multiple of $m$, it holds that*

$$\mathbb{E}[\|\nabla V(\hat{\boldsymbol{s}}^{(K)})\|^2] \leq n^{\frac{2}{3}} \frac{2\overline{L}_{\mathsf{v}}}{\mu K_{\mathsf{max}}} \frac{v_{\max}^2}{v_{\min}^2} \mathbb{E}[V(\hat{\boldsymbol{s}}^{(0)}) - V(\hat{\boldsymbol{s}}^{(K_{\mathsf{max}})})].$$

- *Consider the fiEM method, i.e., Algorithm 1 with (10). Set $\gamma = \frac{v_{\min}}{\alpha \overline{L}_{\mathsf{v}} n^{2/3}}$ such that $\alpha = \max\{6, 1 + 4v_{\min}\}$. For any $K_{\mathsf{max}} \geq 1$, it holds that*

$$\mathbb{E}[\|\nabla V(\hat{\boldsymbol{s}}^{(K)})\|^2] \leq n^{\frac{2}{3}} \frac{\alpha^2 \overline{L}_{\mathsf{v}}}{K_{\mathsf{max}}} \frac{v_{\max}^2}{v_{\min}^2} \mathbb{E}[V(\hat{\boldsymbol{s}}^{(0)}) - V(\hat{\boldsymbol{s}}^{(K_{\mathsf{max}})})].$$

*We recall that $K$ in the above is a uniform and independent r.v. chosen from $[\![K_{\mathsf{max}} - 1]\!]$ [cf. (11)].*

To simplify notation, we shall denote $c_1 = v_{\min}^{-1}$ and $d_1 = v_{\max}$ in the below.

**Proof for the sEM-VR method**  We first establish the following auxiliary lemma:

**Lemma 5.** *For any $k \geq 0$ and consider the update in (9), it holds that*

$$\mathbb{E}[\|\hat{\boldsymbol{s}}^{(k)} - \boldsymbol{\mathcal{S}}^{(k+1)}\|^2] \leq 2\mathbb{E}[\|\hat{\boldsymbol{s}}^{(k)} - \overline{\boldsymbol{s}}^{(k)}\|^2] + 2\,\mathrm{L}_{\mathbf{s}}^2\,\mathbb{E}[\|\hat{\boldsymbol{s}}^{(k)} - \hat{\boldsymbol{s}}^{(\ell(k))}\|^2], \tag{56}$$

*where we recall that $\ell(k) := m\lfloor \frac{k}{m} \rfloor$ is the first iteration number in the epoch that iteration $k$ is in.*

**Proof**  We observe that

$$\mathbb{E}[\|\hat{\boldsymbol{s}}^{(k)} - \boldsymbol{\mathcal{S}}^{(k+1)}\|^2] \leq 2\mathbb{E}[\|\hat{\boldsymbol{s}}^{(k)} - \overline{\boldsymbol{s}}^{(k)}\|^2] + 2\mathbb{E}[\|\overline{\boldsymbol{s}}^{(k)} - \boldsymbol{\mathcal{S}}^{(k+1)}\|^2] \tag{57}$$

For the latter term, we obtain its upper bound as

$$\mathbb{E}[\|\overline{\boldsymbol{s}}^{(k)} - \boldsymbol{\mathcal{S}}^{(k+1)}\|^2] = \mathbb{E}\Big[\Big\|\frac{1}{n}\sum_{i=1}^{n}\big(\overline{\boldsymbol{s}}_i^{(k)} - \overline{\boldsymbol{s}}_i^{\ell(k)}\big) - \big(\overline{\boldsymbol{s}}_{i_k}^{(k)} - \overline{\boldsymbol{s}}_{i_k}^{(\ell(k))}\big)\Big\|^2\Big]$$
$$\leq \mathbb{E}[\|\overline{\boldsymbol{s}}_{i_k}^{(k)} - \overline{\boldsymbol{s}}_{i_k}^{(\ell(k))}\|^2] \leq \mathrm{L}_{\mathbf{s}}^2\,\mathbb{E}[\|\hat{\boldsymbol{s}}^{(k)} - \hat{\boldsymbol{s}}^{(\ell(k))}\|^2] \tag{58}$$

Substituting into (57) proves the lemma.  □

To proceed with our proof, we shall consider a constant step size $\gamma_k = \gamma$ and observe that

$$V(\hat{\boldsymbol{s}}^{(k+1)}) \leq V(\hat{\boldsymbol{s}}^{(k)}) - \gamma\langle \hat{\boldsymbol{s}}^{(k)} - \boldsymbol{\mathcal{S}}^{(k+1)} \,|\, \nabla V(\hat{\boldsymbol{s}}^{(k)})\rangle + \frac{\gamma^2\,\mathrm{L}_V}{2}\|\hat{\boldsymbol{s}}^{(k)} - \boldsymbol{\mathcal{S}}^{(k+1)}\|^2 \tag{59}$$

Using (24) and taking expectations on both sides show that

$$\mathbb{E}[V(\hat{\boldsymbol{s}}^{(k+1)})]$$
$$\leq \mathbb{E}[V(\hat{\boldsymbol{s}}^{(k)})] - \gamma\mathbb{E}\Big[\langle \hat{\boldsymbol{s}}^{(k)} - \overline{\boldsymbol{s}}^{(k)} \,|\, \nabla V(\hat{\boldsymbol{s}}^{(k)})\rangle\Big] + \frac{\gamma^2\,\mathrm{L}_V}{2}\mathbb{E}[\|\hat{\boldsymbol{s}}^{(k)} - \boldsymbol{\mathcal{S}}^{(k+1)}\|^2] \tag{60}$$
$$\overset{(a)}{\leq} \mathbb{E}[V(\hat{\boldsymbol{s}}^{(k)})] - \frac{\gamma}{c_1}\mathbb{E}[\|\hat{\boldsymbol{s}}^{(k)} - \overline{\boldsymbol{s}}^{(k)}\|^2] + \frac{\gamma^2\,\mathrm{L}_V}{2}\mathbb{E}[\|\hat{\boldsymbol{s}}^{(k)} - \boldsymbol{\mathcal{S}}^{(k+1)}\|^2]$$

where (a) is due to Lemma 3. Furthermore, for $k + 1 \leq \ell(k) + m$ (*i.e., $k + 1$ is in the same epoch as $k$*), we have

$$\mathbb{E}[\|\hat{\boldsymbol{s}}^{(k+1)} - \hat{\boldsymbol{s}}^{(\ell(k))}\|^2] = \mathbb{E}[\|\hat{\boldsymbol{s}}^{(k+1)} - \hat{\boldsymbol{s}}^{(k)} + \hat{\boldsymbol{s}}^{(k)} - \hat{\boldsymbol{s}}^{(\ell(k))}\|^2]$$
$$= \mathbb{E}\Big[\|\hat{\boldsymbol{s}}^{(k)} - \hat{\boldsymbol{s}}^{(\ell(k))}\|^2 + \|\hat{\boldsymbol{s}}^{(k+1)} - \hat{\boldsymbol{s}}^{(k)}\|^2 + 2\langle \hat{\boldsymbol{s}}^{(k)} - \hat{\boldsymbol{s}}^{(\ell(k))} \,|\, \hat{\boldsymbol{s}}^{(k+1)} - \hat{\boldsymbol{s}}^{(k)}\rangle\Big]$$
$$= \mathbb{E}\Big[\|\hat{\boldsymbol{s}}^{(k)} - \hat{\boldsymbol{s}}^{(\ell(k))}\|^2 + \gamma^2\|\hat{\boldsymbol{s}}^{(k)} - \boldsymbol{\mathcal{S}}^{(k+1)}\|^2 - 2\gamma\langle \hat{\boldsymbol{s}}^{(k)} - \hat{\boldsymbol{s}}^{(\ell(k))} \,|\, \hat{\boldsymbol{s}}^{(k)} - \overline{\boldsymbol{s}}^{(k)}\rangle\Big] \tag{61}$$
$$\leq \mathbb{E}\Big[(1 + \gamma\beta)\|\hat{\boldsymbol{s}}^{(k)} - \hat{\boldsymbol{s}}^{(\ell(k))}\|^2 + \gamma^2\|\hat{\boldsymbol{s}}^{(k)} - \boldsymbol{\mathcal{S}}^{(k+1)}\|^2 + \frac{\gamma}{\beta}\|\hat{\boldsymbol{s}}^{(k)} - \overline{\boldsymbol{s}}^{(k)}\|^2\Big],$$

where the last inequality is due to the Young's inequality. Consider the following sequence

$$R_k := \mathbb{E}[V(\hat{\boldsymbol{s}}^{(k)}) + b_k \|\hat{\boldsymbol{s}}^{(k)} - \hat{\boldsymbol{s}}^{(\ell(k))}\|^2] \tag{62}$$

where $b_k := \bar{b}_{k \bmod m}$ is a periodic sequence where:

$$\bar{b}_i = \bar{b}_{i+1}(1 + \gamma\beta + 2\gamma^2 \, \mathrm{L}_{\mathbf{s}}^2) + \gamma^2 \, \mathrm{L}_V \, \mathrm{L}_{\mathbf{s}}^2, \quad i = 0, 1, \dots, m-1 \ \ \text{with} \ \bar{b}_m = 0. \tag{63}$$

Note that $\bar{b}_i$ is decreasing with $i$ and this implies

$$\bar{b}_i \leq \bar{b}_0 = \gamma^2 \, \mathrm{L}_V \, \mathrm{L}_{\mathbf{s}}^2 \, \frac{(1 + \gamma\beta + 2\gamma^2 \, \mathrm{L}_{\mathbf{s}}^2)^m - 1}{\gamma\beta + 2\gamma^2 \, \mathrm{L}_{\mathbf{s}}^2}, \quad i = 1, 2, \dots, m. \tag{64}$$

For $k + 1 \leq \ell(k) + m$, we have the following inequality

$$
\begin{aligned}
R_{k+1} &\leq \mathbb{E}\Big[V(\hat{\boldsymbol{s}}^{(k)}) - \frac{\gamma}{c_1}\|\hat{\boldsymbol{s}}^{(k)} - \overline{\boldsymbol{s}}^{(k)}\|^2 + \frac{\gamma^2 \, \mathrm{L}_V}{2}\|\hat{\boldsymbol{s}}^{(k)} - \boldsymbol{\mathcal{S}}^{(k+1)}\|^2\Big] \\
&\quad + b_{k+1}\mathbb{E}\Big[(1 + \gamma\beta)\|\hat{\boldsymbol{s}}^{(k)} - \hat{\boldsymbol{s}}^{(\ell(k))}\|^2 + \gamma^2\|\hat{\boldsymbol{s}}^{(k)} - \boldsymbol{\mathcal{S}}^{(k+1)}\|^2 + \frac{\gamma}{\beta}\|\hat{\boldsymbol{s}}^{(k)} - \overline{\boldsymbol{s}}^{(k)}\|^2\Big] \\
&\overset{(a)}{\leq} \mathbb{E}\Big[V(\hat{\boldsymbol{s}}^{(k)}) - \big(\frac{\gamma}{c_1} - \frac{b_{k+1}\gamma}{\beta}\big)\|\hat{\boldsymbol{s}}^{(k)} - \overline{\boldsymbol{s}}^{(k)}\|^2 + b_{k+1}(1 + \gamma\beta)\|\hat{\boldsymbol{s}}^{(k)} - \hat{\boldsymbol{s}}^{(\ell(k))}\|^2\Big] \\
&\quad + \big(\gamma^2 \, \mathrm{L}_V + 2b_{k+1}\gamma^2\big)\mathbb{E}\Big[\|\hat{\boldsymbol{s}}^{(k)} - \overline{\boldsymbol{s}}^{(k)}\|^2 + \mathrm{L}_{\mathbf{s}}^2\,\|\hat{\boldsymbol{s}}^{(k)} - \hat{\boldsymbol{s}}^{(\ell(k))}\|^2\Big],
\end{aligned}
\tag{65}
$$

where (a) is due to Lemma 5. Rearranging terms gives

$$
\begin{aligned}
R_{k+1} &\leq \mathbb{E}[V(\hat{\boldsymbol{s}}^{(k)})] - \big(\frac{\gamma}{c_1} - \frac{b_{k+1}\gamma}{\beta} - \gamma^2(\mathrm{L}_V + 2b_{k+1})\big)\mathbb{E}[\|\hat{\boldsymbol{s}}^{(k)} - \overline{\boldsymbol{s}}^{(k)}\|^2] \\
&\quad + \Big(\underbrace{b_{k+1}(1 + \gamma\beta + 2\gamma^2 \, \mathrm{L}_{\mathbf{s}}^2) + \gamma^2 \, \mathrm{L}_V \, \mathrm{L}_{\mathbf{s}}^2}_{=b_k \ \text{since } k+1 \leq \ell(k)+m}\Big)\mathbb{E}\Big[\|\hat{\boldsymbol{s}}^{(k)} - \hat{\boldsymbol{s}}^{(\ell(k))}\|^2\Big] \\
&= R_k - \big(\frac{\gamma}{c_1} - \frac{b_{k+1}\gamma}{\beta} - \gamma^2(\mathrm{L}_V + 2b_{k+1})\big)\mathbb{E}[\|\hat{\boldsymbol{s}}^{(k)} - \overline{\boldsymbol{s}}^{(k)}\|^2]
\end{aligned}
\tag{66}
$$

This leads to, for any $\gamma$ and $\beta$ such that $(1 - c_1 b_{k+1}\beta^{-1} - c_1\gamma(\mathrm{L}_V + 2b_{k+1}) > 0$,

$$\frac{1}{d_1^2}\mathbb{E}[\|\nabla V(\hat{\boldsymbol{s}}^{(k)})\|^2] \leq \mathbb{E}[\|\hat{\boldsymbol{s}}^{(k)} - \overline{\boldsymbol{s}}^{(k)}\|^2] \leq \frac{c_1(R_k - R_{k+1})}{\gamma\big(1 - c_1 b_{k+1}\beta^{-1} - c_1\gamma(\mathrm{L}_V + 2b_{k+1})\big)}. \tag{67}$$

By setting $\beta = \frac{c_1 \overline{L}_v}{n^{1/3}}$, $\gamma = \frac{\mu}{c_1 \overline{L}_v n^{2/3}}$, $m = \frac{nc_1^2}{2\mu^2 + \mu c_1^2}$, it can be shown that there exists $\mu \in (0, 1)$, such that the following lower bound holds

$$
\begin{aligned}
1 - c_1\gamma \, \mathrm{L}_V - \big(\frac{c_1}{\beta} + 2c_1\gamma\big)b_{k+1} &\geq 1 - \frac{\mu}{n^{\frac{2}{3}}} - \bar{b}_0\big(\frac{n^{\frac{1}{3}}}{\overline{L}_v} + \frac{2\mu}{\overline{L}_v n^{\frac{2}{3}}}\big) \\
&\geq 1 - \frac{\mu}{n^{\frac{2}{3}}} - \frac{\mathrm{L}_V \, \mu^2}{c_1^2 n^{\frac{4}{3}}}\frac{(1 + \gamma\beta + 2\gamma^2 \, \mathrm{L}_{\mathbf{s}}^2)^m - 1}{\gamma\beta + 2\gamma^2 \, \mathrm{L}_{\mathbf{s}}^2}\big(\frac{n^{\frac{1}{3}}}{\overline{L}_v} + \frac{2\mu}{\overline{L}_v n^{\frac{2}{3}}}\big) \\
&\overset{(a)}{\geq} 1 - \frac{\mu}{n^{\frac{2}{3}}} - \frac{\mu}{c_1^2}(e - 1)\big(1 + \frac{2\mu}{n}\big) \geq 1 - \mu - \mu(1 + 2\mu)\frac{e - 1}{c_1^2} \overset{(b)}{\geq} \frac{1}{2}
\end{aligned}
\tag{68}
$$

where the simplification in (a) is due to

$$\frac{\mu}{n} \leq \gamma\beta + 2\gamma^2 \, \mathrm{L}_{\mathbf{s}}^2 \leq \frac{\mu}{n} + \frac{2\mu^2}{c_1^2 n^{\frac{4}{3}}} \leq \frac{\mu c_1^2 + 2\mu^2}{c_1^2}\frac{1}{n} \quad \text{and} \quad (1 + \gamma\beta + 2\gamma^2 \, \mathrm{L}_{\mathbf{s}}^2)^m \leq e - 1. \tag{69}$$

and the required $\mu$ in (b) can be found by solving the quadratic equation[1]. This gives

$$\mathbb{E}[\|\nabla V(\hat{\boldsymbol{s}}^{(K)})\|^2] = \frac{1}{K_{\max}}\sum_{k=0}^{K_{\max}-1}\mathbb{E}[\|\nabla V(\hat{\boldsymbol{s}}^{(k)})\|^2] \leq \frac{2d_1^2 c_1(R_0 - R_{K_{\max}})}{\gamma K_{\max}} \tag{70}$$

Note that $R_0 = \mathbb{E}[V(\hat{\boldsymbol{s}}^{(0)})]$ and if $K_{\max}$ is a multiple of $m$, then $R_{\max} = \mathbb{E}[V(\hat{\boldsymbol{s}}^{(K_{\max})})]$. Under the latter condition, we have

$$\mathbb{E}[\|\nabla V(\hat{\boldsymbol{s}}^{(K)})\|^2] \leq n^{\frac{2}{3}}\frac{2d_1^2 c_1^2 \overline{L}_v}{\mu K_{\max}}\mathbb{E}[V(\hat{\boldsymbol{s}}^{(0)}) - V(\hat{\boldsymbol{s}}^{(K_{\max})})]. \tag{71}$$

This concludes our proof.

**Proof for the fiEM method**   Our proof proceeds by observing the following auxiliary lemma:

**Lemma 6.** *For any $k \geq 0$ and consider the update in* (10)*, it holds that*

$$\mathbb{E}[\|\hat{s}^{(k)} - \boldsymbol{\mathcal{S}}^{(k+1)}\|^2] \leq 2\mathbb{E}[\|\hat{s}^{(k)} - \overline{s}^{(k)}\|^2] + \frac{2\,\mathrm{L}_\mathbf{s}^2}{n}\sum_{i=1}^{n}\mathbb{E}[\|\hat{s}^{(k)} - \hat{s}^{(t_i^k)}\|^2] \tag{72}$$

**Proof**   We observe that $\overline{\boldsymbol{\mathcal{S}}}^{(k)} = \frac{1}{n}\sum_{i=1}^{n}\overline{s}_i^{(t_i^k)}$ and $\mathbb{E}[\overline{s}_{i_k}^{(k)} - \overline{s}_{i_k}^{(t_{i_k}^k)}] = \overline{s}^{(k)} - \overline{\boldsymbol{\mathcal{S}}}^{(k)}$. Moreover, we recall that $\overline{s}_i^{(k)} = \overline{s}_i(\hat{\boldsymbol{\theta}}^{(k)}) = \overline{s}_i(\overline{\boldsymbol{\theta}}(\hat{s}^{(k)}))$. Thus

$$\mathbb{E}[\|\hat{s}^{(k)} - \boldsymbol{\mathcal{S}}^{(k+1)}\|^2] \overset{(a)}{=} \mathbb{E}[\|\hat{s}^{(k)} - \overline{s}^{(k)} + (\overline{s}^{(k)} - \overline{\boldsymbol{\mathcal{S}}}^{(k)}) - (\overline{s}_{i_k}^{(k)} - \overline{s}_{i_k}^{(t_{i_k}^k)})\|^2]$$

$$\leq 2\mathbb{E}[\|\hat{s}^{(k)} - \overline{s}^{(k)}\|^2] + 2\mathbb{E}[\|(\overline{s}^{(k)} - \overline{\boldsymbol{\mathcal{S}}}^{(k)}) - (\overline{s}_{i_k}^{(k)} - \overline{s}_{i_k}^{(t_{i_k}^k)})\|^2] \tag{73}$$

$$\overset{(b)}{\leq} 2\mathbb{E}[\|\hat{s}^{(k)} - \overline{s}^{(k)}\|^2] + 2\mathbb{E}[\|\overline{s}_{i_k}^{(k)} - \overline{s}_{i_k}^{(t_{i_k}^k)}\|^2],$$

where (a) uses the SAGA update in (10); (b) uses the variance inequality $\mathbb{E}[\|X - \mathbb{E}[X]\|^2] \leq \mathbb{E}[\|X\|^2]$. The last expectation can be further bounded by

$$\mathbb{E}[\|\overline{s}_{i_k}^{(k)} - \overline{s}_{i_k}^{(t_{i_k}^k)}\|^2] = \frac{1}{n}\sum_{i=1}^{n}\mathbb{E}[\|\overline{s}_i^{(k)} - \overline{s}_i^{(t_i^k)}\|^2] \overset{(a)}{\leq} \frac{\mathrm{L}_\mathbf{s}}{n}\sum_{i=1}^{n}\mathbb{E}[\|\hat{s}^{(k)} - \hat{s}^{(t_i^k)}\|^2], \tag{74}$$

where (a) is due to Lemma 3. Combining the two equations above yields the desired lemma.   $\square$

Let $\gamma_{k+1} = \gamma$, *i.e.*, with a fixed step size. We observe the following

$$V(\hat{s}^{(k+1)}) \leq V(\hat{s}^{(k)}) - \gamma\langle \hat{s}^{(k)} - \boldsymbol{\mathcal{S}}^{(k+1)} \mid \nabla V(\hat{s}^{(k)})\rangle + \frac{\gamma^2\,\mathrm{L}_V}{2}\|\hat{s}^{(k)} - \boldsymbol{\mathcal{S}}^{(k+1)}\|^2 \tag{75}$$

Taking expectations on both sides yields

$$\mathbb{E}[V(\hat{s}^{(k+1)})]$$

$$\leq \mathbb{E}[V(\hat{s}^{(k)})] - \gamma\mathbb{E}\Big[\langle \hat{s}^{(k)} - \overline{s}^{(k)} \mid \nabla V(\hat{s}^{(k)})\rangle\Big] + \frac{\gamma^2\,\mathrm{L}_V}{2}\mathbb{E}[\|\hat{s}^{(k)} - \boldsymbol{\mathcal{S}}^{(k+1)}\|^2]$$

$$\overset{(a)}{\leq} \mathbb{E}[V(\hat{s}^{(k)})] - \frac{\gamma}{c_1}\mathbb{E}[\|\hat{s}^{(k)} - \overline{s}^{(k)}\|^2] + \frac{\gamma^2\,\mathrm{L}_V}{2}\mathbb{E}[\|\hat{s}^{(k)} - \boldsymbol{\mathcal{S}}^{(k+1)}\|^2] \tag{76}$$

$$\overset{(b)}{\leq} \mathbb{E}[V(\hat{s}^{(k)})] - \Big(\frac{\gamma}{c_1} - \gamma^2\,\mathrm{L}_V\Big)\mathbb{E}[\|\hat{s}^{(k)} - \overline{s}^{(k)}\|^2] + \frac{\gamma^2\,\mathrm{L}_V\,\mathrm{L}_\mathbf{s}^2}{n}\sum_{i=1}^{n}\mathbb{E}[\|\hat{s}^{(k)} - \hat{s}^{(t_i^k)}\|^2]$$

where (a) is due to Lemma 3 and (b) is due to Lemma 6. Next, we observe that

$$\frac{1}{n}\sum_{i=1}^{n}\mathbb{E}[\|\hat{s}^{(k+1)} - \hat{s}^{(t_i^{k+1})}\|^2] = \frac{1}{n}\sum_{i=1}^{n}\Big(\frac{1}{n}\mathbb{E}[\|\hat{s}^{(k+1)} - \hat{s}^{(k)}\|^2] + \frac{n-1}{n}\mathbb{E}[\|\hat{s}^{(k+1)} - \hat{s}^{(t_i^k)}\|^2]\Big) \tag{77}$$

where the equality holds as $i_k$ and $j_k$ are drawn independently. For any $\beta > 0$, it holds

$$\mathbb{E}[\|\hat{s}^{(k+1)} - \hat{s}^{(t_i^k)}\|^2]$$

$$= \mathbb{E}\Big[\|\hat{s}^{(k+1)} - \hat{s}^{(k)}\|^2 + \|\hat{s}^{(k)} - \hat{s}^{(t_i^k)}\|^2 + 2\langle \hat{s}^{(k+1)} - \hat{s}^{(k)} \mid \hat{s}^{(k)} - \hat{s}^{(t_i^k)}\rangle\Big]$$

$$= \mathbb{E}\Big[\|\hat{s}^{(k+1)} - \hat{s}^{(k)}\|^2 + \|\hat{s}^{(k)} - \hat{s}^{(t_i^k)}\|^2 - 2\gamma\langle \hat{s}^{(k)} - \overline{s}^{(k)} \mid \hat{s}^{(k)} - \hat{s}^{(t_i^k)}\rangle\Big] \tag{78}$$

$$\leq \mathbb{E}\Big[\|\hat{s}^{(k+1)} - \hat{s}^{(k)}\|^2 + \|\hat{s}^{(k)} - \hat{s}^{(t_i^k)}\|^2 + \frac{\gamma}{\beta}\|\hat{s}^{(k)} - \overline{s}^{(k)}\|^2 + \gamma\beta\|\hat{s}^{(k)} - \hat{s}^{(t_i^k)}\|^2\Big]$$

where the last inequality is due to the Young's inequality. Subsequently, we have

$$\frac{1}{n}\sum_{i=1}^{n}\mathbb{E}[\|\hat{s}^{(k+1)} - \hat{s}^{(t_i^{k+1})}\|^2]$$

$$\leq \mathbb{E}[\|\hat{s}^{(k+1)} - \hat{s}^{(k)}\|^2] + \frac{n-1}{n^2}\sum_{i=1}^{n}\mathbb{E}\Big[(1+\gamma\beta)\|\hat{s}^{(k)} - \hat{s}^{(t_i^k)}\|^2 + \frac{\gamma}{\beta}\|\hat{s}^{(k)} - \overline{s}^{(k)}\|^2\Big] \tag{79}$$

Observe that $\hat{s}^{(k+1)} - \hat{s}^{(k)} = -\gamma(\hat{s}^{(k)} - \mathcal{S}^{(k+1)})$. Applying Lemma 6 yields

$$
\frac{1}{n}\sum_{i=1}^{n}\mathbb{E}[\|\hat{s}^{(k+1)} - \hat{s}^{(t_i^{k+1})}\|^2]
$$

$$
\leq \left(2\gamma^2 + \frac{n-1}{n}\frac{\gamma}{\beta}\right)\mathbb{E}[\|\hat{s}^{(k)} - \overline{s}^{(k)}\|^2] + \sum_{i=1}^{n}\left(\frac{2\gamma^2\,L_s^2}{n} + \frac{(n-1)(1+\gamma\beta)}{n^2}\right)\mathbb{E}[\|\hat{s}^{(k)} - \hat{s}^{(t_i^k)}\|^2]
$$

$$
\leq \left(2\gamma^2 + \frac{\gamma}{\beta}\right)\mathbb{E}[\|\hat{s}^{(k)} - \overline{s}^{(k)}\|^2] + \sum_{i=1}^{n}\frac{1 - \frac{1}{n} + \gamma\beta + 2\gamma^2\,L_s^2}{n}\mathbb{E}[\|\hat{s}^{(k)} - \hat{s}^{(t_i^k)}\|^2]
$$

Let us define

$$
\Delta^{(k)} := \frac{1}{n}\sum_{i=1}^{n}\mathbb{E}[\|\hat{s}^{(k)} - \hat{s}^{(t_i^k)}\|^2] \tag{80}
$$

From the above, we get

$$
\Delta^{(k+1)} \leq \left(1 - \frac{1}{n} + \gamma\beta + 2\gamma^2\,L_s^2\right)\Delta^{(k)} + \left(2\gamma^2 + \frac{\gamma}{\beta}\right)\mathbb{E}[\|\hat{s}^{(k)} - \overline{s}^{(k)}\|^2] \tag{81}
$$

Setting $\overline{L}_v = \max\{L_s, L_V\}$, $\gamma = \frac{1}{\alpha c_1 \overline{L}_v n^{2/3}}$, $\beta = \frac{c_1 \overline{L}_v}{n^{1/3}}$, $(\alpha-1)c_1 \geq 4$, $\alpha \geq 6$, it is easy to check that

$$
1 - \frac{1}{n} + \gamma\beta + 2\gamma^2\,L_s^2 \geq 1 - \frac{1}{n} \tag{82}
$$

and

$$
1 - \frac{1}{n} + \gamma\beta + 2\gamma^2\,L_s^2 \leq 1 - \frac{1}{n} + \frac{1}{\alpha n} + \frac{2}{\alpha^2 c_1^2 n^{\frac{4}{3}}} \leq 1 - \frac{\alpha c_1 - c_1 - 2}{\alpha c_1 n} \leq 1 - \frac{2}{\alpha c_1 n} \tag{83}
$$

which shows that $1 - \frac{1}{n} + \gamma\beta + 2\gamma^2\,L_s^2 \in (0,1)$. Observe that as $\Delta^{(0)} = 0$ and by telescoping, we have

$$
\Delta^{(k+1)} \leq \left(2\gamma^2 + \frac{\gamma}{\beta}\right)\sum_{\ell=0}^{k}\left(1 - \frac{1}{n} + \gamma\beta + 2\gamma^2\,L_s^2\right)^{k-\ell}\mathbb{E}[\|\hat{s}^{(\ell)} - \overline{s}^{(\ell)}\|^2] \tag{84}
$$

Let $K_{\max} \in \mathbb{N}$. Summing $k = 0$ to $k = K_{\max} - 1$ gives

$$
\sum_{k=0}^{K_{\max}-1}\Delta^{(k+1)} \leq \left(2\gamma^2 + \frac{\gamma}{\beta}\right)\sum_{k=0}^{K_{\max}-1}\sum_{\ell=0}^{k}\left(1 - \frac{1}{n} + \gamma\beta + 2\gamma^2\,L_s^2\right)^{k-\ell}\mathbb{E}[\|\hat{s}^{(\ell)} - \overline{s}^{(\ell)}\|^2]
$$

$$
= \left(2\gamma^2 + \frac{\gamma}{\beta}\right)\sum_{k=0}^{K_{\max}-1}\sum_{\ell=0}^{k}\left(1 - \frac{1}{n} + \gamma\beta + 2\gamma^2\,L_s^2\right)^{\ell}\mathbb{E}[\|\hat{s}^{(k)} - \overline{s}^{(k)}\|^2] \tag{85}
$$

$$
\leq \frac{2\gamma^2 + \frac{\gamma}{\beta}}{\frac{1}{n} - \gamma\beta - 2\gamma^2 L_s^2}\sum_{k=0}^{K_{\max}-1}\mathbb{E}[\|\hat{s}^{(k)} - \overline{s}^{(k)}\|^2].
$$

Summing up the both sides of (76) from $k = 0$ to $k = K_{\max} - 1$ yields

$$
\mathbb{E}\left[V(\hat{s}^{(K_{\max})}) - V(\hat{s}^{(0)})\right]
$$

$$
\leq \sum_{k=0}^{K_{\max}-1}\left\{\left(-\frac{\gamma}{c_1} + \gamma^2\,L_V\right)\mathbb{E}[\|\hat{s}^{(k)} - \overline{s}^{(k)}\|^2] + \gamma^2\,L_V\,L_s^2\,\Delta^{(k)}\right\} \tag{86}
$$

$$
\leq \sum_{k=0}^{K_{\max}-1}\left\{\left(-\frac{\gamma}{c_1} + \gamma^2\,L_V + \frac{(\gamma^2\,L_V\,L_s^2)(2\gamma^2 + \frac{\gamma}{\beta})}{\frac{1}{n} - \gamma\beta - 2\gamma^2 L_s^2}\right)\mathbb{E}[\|\hat{s}^{(k)} - \overline{s}^{(k)}\|^2]\right\}
$$

Furthermore,

$$\gamma^2 \, \mathrm{L}_V + \frac{\left(\gamma^2 \, \mathrm{L}_V \, \mathrm{L}_{\mathbf{s}}^2\right)\left(2\gamma^2 + \frac{\gamma}{\beta}\right)}{\frac{1}{n} - \gamma\beta - 2\gamma^2 L_{\mathbf{s}}^2}$$

$$\overset{(a)}{\leq} \frac{1}{\alpha^2 c_1^2 \overline{L}_{\mathsf{v}} n^{4/3}} + \frac{\overline{L}_{\mathsf{v}}(\alpha^2 c_1^2 n^{4/3})^{-1}\left(\frac{2}{\alpha^2 c_1^2 \overline{L}_{\mathsf{v}}^2 n^{4/3}} + \frac{1}{\alpha c_1^2 \overline{L}_{\mathsf{v}}^2 n^{1/3}}\right)}{\frac{1}{n} - \frac{1}{\alpha n} - \frac{2}{\alpha^2 c_1^2 n^{4/3}}}$$

$$= \frac{1}{\alpha^2 c_1^2 \overline{L}_{\mathsf{v}} n^{4/3}} + \frac{\overline{L}_{\mathsf{v}}\left(\frac{2}{\alpha^2 c_1^2 \overline{L}_{\mathsf{v}}^2 n^{4/3}} + \frac{1}{\alpha c_1^2 \overline{L}_{\mathsf{v}}^2 n^{1/3}}\right)}{(\alpha c_1 n^{1/3})(\alpha - 1)c_1 - 2} \qquad (87)$$

$$\overset{(b)}{\leq} \frac{1}{\alpha^2 c_1^2 \overline{L}_{\mathsf{v}} n^{4/3}} + \frac{\frac{1}{\alpha c_1^2 \overline{L}_{\mathsf{v}} n^{1/3}}\left(\frac{2}{\alpha n} + 1\right)}{4(\alpha c_1 n^{1/3}) - 2} \overset{(c)}{\leq} \frac{1}{\alpha^2 c_1^2 \overline{L}_{\mathsf{v}} n^{4/3}} + \frac{2}{3\alpha^2 c_1^3 \overline{L}_{\mathsf{v}} n^{2/3}}$$

$$\leq \frac{5/6}{\alpha c_1^2 \overline{L}_{\mathsf{v}} n^{2/3}}$$

where (a) uses $\overline{L}_{\mathsf{v}} \geq \max\{\mathrm{L}_{\mathbf{s}}, \mathrm{L}_V\}$, (b) is due to $(\alpha - 1)c_1 \geq 4$ and (c) uses $\alpha c_1 n^{1/3} \geq 1$. Now, using the fact that $\frac{\gamma}{c_1} = \frac{1}{\alpha c_1^2 \overline{L}_{\mathsf{v}} n^{\frac{2}{3}}}$ and the lower bound $\|\hat{\mathbf{s}}^{(k)} - \overline{\mathbf{s}}^{(k)}\|^2 \geq d_2^{-1}\|\nabla V(\hat{\mathbf{s}}^{(k)})\|^2$, we have

$$\mathbb{E}\big[V(\hat{\mathbf{s}}^{(K_{\max})}) - V(\hat{\mathbf{s}}^{(0)})\big] \leq -\frac{1}{6\alpha c_1^2 \overline{L}_{\mathsf{v}} n^{\frac{2}{3}}} \sum_{k=0}^{K_{\max}-1} \mathbb{E}[\|\hat{\mathbf{s}}^{(k)} - \overline{\mathbf{s}}^{(k)}\|^2]$$

$$\leq -\frac{1}{6\alpha d_1^2 c_1^2 \overline{L}_{\mathsf{v}} n^{\frac{2}{3}}} \sum_{k=0}^{K_{\max}-1} \mathbb{E}[\|\nabla V(\hat{\mathbf{s}}^{(k)})\|^2] \qquad (88)$$

Recalling that $K$ is an independent discrete r.v. drawn uniformly from $\{1, \ldots, K_{\max}\}$ and noting that $\alpha \geq 6$, we have

$$\mathbb{E}[\|\nabla V(\hat{\mathbf{s}}^{(K)})\|^2] = \frac{1}{K_{\max}} \sum_{k=0}^{K_{\max}-1} \mathbb{E}[\|\nabla V(\hat{\mathbf{s}}^{(k)})\|^2] \leq n^{\frac{2}{3}} \frac{d_1^2 \overline{L}_{\mathsf{v}}(\alpha c_1)^2(\mathbb{E}\big[V(\hat{\mathbf{s}}^{(0)}) - V(\hat{\mathbf{s}}^{(K_{\max})})\big]}{K_{\max}}$$

$$(89)$$

# G  Practical Applications of Stochastic EM methods

This section provides implementation details and verify the model assumptions for the application examples provided. Only in this section, for any $M \geq 2$, we denote

$$\Delta^M := \{\omega_m \in \mathbb{R}, \ m = 1, \ldots, M-1 : \omega_m \geq 0, \ \textstyle\sum_{m=1}^{M-1} \omega_m \leq 1\} \subseteq \mathbb{R}^{M-1} \qquad (90)$$

as the shorthand notation of the dimension reduced $M$-D probability simplex.

## G.1  Gaussian mixture models

### G.1.1  Model assumptions

We first recognize that the constraint set for $\boldsymbol{\theta}$ is given by

$$\Theta = \Delta^M \times \mathbb{R}^M. \qquad (91)$$

Using the partition of the sufficient statistics as $S(y_i, z_i) = (S^{(1)}(y_i, z_i)^\top, S^{(2)}(y_i, z_i)^\top, S^{(3)}(y_i, z_i))^\top \in \mathbb{R}^{M-1} \times \mathbb{R}^{M-1} \times \mathbb{R}$, the partition $\phi(\boldsymbol{\theta}) = (\phi^{(1)}(\boldsymbol{\theta})^\top, \phi^{(2)}(\boldsymbol{\theta})^\top, \phi^{(3)}(\boldsymbol{\theta}))^\top \in \mathbb{R}^{M-1} \times \mathbb{R}^{M-1} \times \mathbb{R}$ and the fact that $\mathbb{1}_{\{M\}}(z_i) = 1 - \sum_{m=1}^{M-1} \mathbb{1}_{\{m\}}(z_i)$, the complete data log-likelihood can be expressed as in (2) with

$$s_{i,m}^{(1)} = \mathbb{1}_{\{m\}}(z_i), \quad \phi_m^{(1)}(\boldsymbol{\theta}) = \left\{\log(\omega_m) - \frac{\mu_m^2}{2}\right\} - \left\{\log(1 - \textstyle\sum_{j=1}^{M-1}\omega_j) - \frac{\mu_M^2}{2}\right\},$$

$$s_{i,m}^{(2)} = \mathbb{1}_{\{m\}}(z_i)y_i, \quad \phi_m^{(2)}(\boldsymbol{\theta}) = \mu_m, \quad s_i^{(3)} = y_i, \quad \phi^{(3)}(\boldsymbol{\theta}) = \mu_M, \qquad (92)$$

and $\psi(\boldsymbol{\theta}) = -\left\{\log(1 - \sum_{m=1}^{M-1} \omega_m) - \frac{\mu_M^2}{2\sigma^2}\right\}$. We also define for each $m \in [\![1, M]\!]$, $j \in [\![1, 3]\!]$, $s_m^{(j)} = n^{-1} \sum_{i=1}^{n} s_{i,m}^{(j)}$. Consider the following conditional expected value:

$$\widetilde{\omega}_m(y_i; \boldsymbol{\theta}) := \mathbb{E}_{\boldsymbol{\theta}}[\mathbb{1}_{\{z_i=m\}} | y = y_i] = \frac{\omega_m \exp(-\frac{1}{2}(y_i - \mu_i)^2)}{\sum_{j=1}^{M} \omega_j \exp(-\frac{1}{2}(y_i - \mu_j)^2)} , \tag{93}$$

where $m \in [\![1, M]\!]$, $i \in [\![1, n]\!]$ and $\boldsymbol{\theta} = (\boldsymbol{w}, \boldsymbol{\mu}) \in \Theta$. In particular, given $\boldsymbol{\theta} \in \Theta$, the E-step updates in (6) can be written as

$$\mathbf{\bar{s}}_i(\boldsymbol{\theta}) = \big(\underbrace{\widetilde{\omega}_1(y_i; \boldsymbol{\theta}), ..., \widetilde{\omega}_{M-1}(y_i; \boldsymbol{\theta})}_{:=\mathbf{\bar{s}}_i^{(1)}(\boldsymbol{\theta})^\top}, \underbrace{y_i\widetilde{\omega}_1(y_i; \boldsymbol{\theta}), ..., y_i\widetilde{\omega}_M(y_i; \boldsymbol{\theta})}_{:=\mathbf{\bar{s}}_i^{(2)}(\boldsymbol{\theta})^\top}, \underbrace{y_i}_{:=\mathbf{\bar{s}}_i^{(3)}(\boldsymbol{\theta})} \big)^\top. \tag{94}$$

Recall that we have used the following regularizer:

$$\mathrm{R}(\boldsymbol{\theta}) = \frac{\delta}{2} \sum_{m=1}^{M} \mu_m^2 - \epsilon \sum_{m=1}^{M} \log(\omega_m) - \epsilon \log\big(1 - \sum_{m=1}^{M-1} \omega_m\big) , \tag{95}$$

It can be shown that the regularized M-step in (5) evaluates to

$$\overline{\boldsymbol{\theta}}(\boldsymbol{s}) = \begin{pmatrix} (1 + \epsilon M)^{-1}\big(s_1^{(1)} + \epsilon, \dots, s_{M-1}^{(1)} + \epsilon\big)^\top \\ \big((s_1^{(1)} + \delta)^{-1} s_1^{(2)}, \dots, (s_{M-1}^{(1)} + \delta)^{-1} s_{M-1}^{(2)}\big)^\top \\ \big(1 - \sum_{m=1}^{M-1} s_m^{(1)} + \delta\big)^{-1}\big(s^{(3)} - \sum_{m=1}^{M-1} s_m^{(2)}\big) \end{pmatrix} = \begin{pmatrix} \overline{\boldsymbol{\omega}}(\boldsymbol{s}) \\ \overline{\boldsymbol{\mu}}(\boldsymbol{s}) \\ \overline{\mu}_M(\boldsymbol{s}) \end{pmatrix} . \tag{96}$$

where we have defined for all $m \in [\![1, M]\!]$ and $j \in [\![1, 3]\!]$, $s_m^{(j)} = n^{-1} \sum_{i=1}^{n} s_{i,m}^{(j)}$.

To analyze the convergence of the EM methods, we verify H1 to H5 for the GMM example as follows.

To verify H1, we observe that the set Z is the compact interval $[\![M]\!]$, in addition, the sufficient statistics defined in (92) also leads to a bounded and closed S.

To verify H2, we observe that the Jacobian matrix $\mathrm{J}_\phi^{\boldsymbol{\theta}}(\boldsymbol{\theta})$ can be computed as

$$\mathrm{J}_\phi^{\boldsymbol{\theta}}(\boldsymbol{\theta}) = \begin{pmatrix} \frac{1}{1 - \sum_{m=1}^{M-1} \omega_m} \mathbf{1}\mathbf{1}^\top + \mathrm{Diag}(\frac{\mathbf{1}}{\boldsymbol{\omega}}) & -\mathrm{Diag}(\boldsymbol{\mu}) & \mu_M \mathbf{1} \\ \mathbf{0} & \boldsymbol{I} & \mathbf{0} \\ \mathbf{0} & \mathbf{0} & 1 \end{pmatrix} , \tag{97}$$

where we have denoted $\frac{\mathbf{1}}{\boldsymbol{\omega}}$ as the $(M-1)$-dimensional vector $\big(\frac{1}{\omega_1}, \dots, \frac{1}{\omega_{M-1}}\big)$. We observe that it is a bounded matrix and it is smooth *w.r.t.* $\boldsymbol{\theta}$.

We verify H3 next, *i.e.,* the Lipschitz continuity of $p(z_i|y_i; \boldsymbol{\theta})$, w.r.t to $\boldsymbol{\theta}$ noting that for all $i \in [\![n]\!]$ and $m \in [\![M]\!]$, $p(z_i = m|y_i; \boldsymbol{\theta}) = \mathbb{E}_{\boldsymbol{\theta}}[\mathbb{1}_{\{z_i=m\}}|y = y_i] = \widetilde{\omega}_m(y_i; \boldsymbol{\theta})$. Observe that $p(z_i = m|y_i; \boldsymbol{\theta})$ is given by the softmax function and the desired Lipschitz property follows.

Next, we observe that with the designed penalty, the function $\boldsymbol{\theta} \mapsto L(\mathbf{s}, \boldsymbol{\theta})$ admits a unique global minima with $\overline{\boldsymbol{\theta}}(\mathbf{s}) \in \mathrm{int}(\Theta)$ for all $\mathbf{s} \in \mathsf{S}$. Second, since $\overline{\boldsymbol{\theta}}(\mathbf{s}) \in \mathrm{int}(\Theta)$, the Jacobian matrix defined in (97) must be full rank. Lastly, the $L_\theta$-Lipschitzness of $\overline{\boldsymbol{\theta}}(\mathbf{s})$ can be deduced by inspecting (96). The above show that Assumption H4 is verified.

Finally, we calculate the quantity $\mathrm{B}(\mathbf{s})$ defined in (17). Observe that the Hessian $\mathrm{H}_L^{\boldsymbol{\theta}}(\mathbf{s}, \boldsymbol{\theta})$ is:

$$\mathrm{H}_L^{\boldsymbol{\theta}}(\mathbf{s}, \boldsymbol{\theta}) = \begin{pmatrix} \frac{1 + \epsilon - \sum_{m=1}^{M-1} s_m^{(1)}}{(1 - \sum_{m=1}^{M-1} \omega_m)^2} \mathbf{1}\mathbf{1}^\top + \mathrm{Diag}(\frac{\boldsymbol{s}^{(1)} + \epsilon\mathbf{1}}{\boldsymbol{\omega}^2}) & \mathbf{0} & \mathbf{0} \\ \mathbf{0} & \mathrm{Diag}(\boldsymbol{s}^{(1)} + \delta\mathbf{1}) & \mathbf{0} \\ \mathbf{0} & \mathbf{0} & \delta + 1 - \sum_{m=1}^{M-1} s_m^{(1)} \end{pmatrix} \tag{98}$$

We can rewrite $\mathrm{B}(\mathbf{s})$ as an outer product:

$$\mathrm{B}(\mathbf{s}) := \mathrm{J}_\phi^{\boldsymbol{\theta}}(\overline{\boldsymbol{\theta}}(\mathbf{s}))\big(\mathrm{H}_L^{\boldsymbol{\theta}}(\mathbf{s}, \overline{\boldsymbol{\theta}}(\mathbf{s}))\big)^{-1} \mathrm{J}_\phi^{\boldsymbol{\theta}}(\overline{\boldsymbol{\theta}}(\mathbf{s}))^\top = \boldsymbol{\mathcal{J}}(\boldsymbol{s})\boldsymbol{\mathcal{J}}(\boldsymbol{s})^\top \tag{99}$$

where

$$\mathcal{J}(s) := \mathrm{J}_\phi^{\boldsymbol{\theta}}(\overline{\boldsymbol{\theta}}(s)) \begin{pmatrix} \boldsymbol{H}_{11}^{-\frac{1}{2}} & \mathbf{0} & \mathbf{0} \\ \mathbf{0} & \mathrm{Diag}(\frac{\mathbf{1}}{\sqrt{s^{(1)}+\delta\mathbf{1}}}) & \mathbf{0} \\ \mathbf{0} & \mathbf{0} & \frac{1}{\sqrt{\delta+1-\sum_{m=1}^{M-1} s_m^{(1)}}} \end{pmatrix} \tag{100}$$

and

$$\boldsymbol{H}_{11} := \frac{1+\epsilon-\sum_{m=1}^{M-1} s_m^{(1)}}{(1-\frac{\mathbf{1}^\top (s^{(1)}+\epsilon\mathbf{1})}{1+\epsilon M})^2}\mathbf{1}\mathbf{1}^\top + \mathrm{Diag}(\frac{(1+\epsilon M)^2}{s^{(1)}+\epsilon\mathbf{1}}). \tag{101}$$

Note that $\mathcal{J}(s)$ is a bounded and full rank matrix which yields to the upper and lower bounds on eigenvealues in H5. From (100), we note that $\mathrm{B}(s) = \mathcal{J}(s)\mathcal{J}(s)^\top$ is Lipschitz continuous, *i.e.,* , there exists a constant $\mathrm{L}_B$ such that for all $s, s' \in \mathsf{S}^2$, we have $\| \mathrm{B}(s) - \mathrm{B}(s') \| \leq \mathrm{L}_B \| s - s' \|$.

### G.1.2   Algorithms updates

In the sequel, for all $i \in [\![n]\!]$ and iteration $k$, the conditional expectation $\overline{\mathbf{s}}_i^{(k)}$ is defined by (94) and is equal to:

$$\overline{\mathbf{s}}_i^{(k)} = \begin{pmatrix} \left(\widetilde{\omega}_1(y_i;\hat{\boldsymbol{\theta}}^{(k)}),\dots,\widetilde{\omega}_{M-1}(y_i;\hat{\boldsymbol{\theta}}^{(k)})\right)^\top \\ \left(y_i\widetilde{\omega}_1(y_i;\hat{\boldsymbol{\theta}}^{(k)}),\dots,y_i\widetilde{\omega}_{M-1}(y_i;\hat{\boldsymbol{\theta}}^{(k)})\right)^\top \\ y_i \end{pmatrix}. \tag{102}$$

At iteration $k$, the several E-steps defined by (7) or (8) or (9) or (10) leads to the definition of the quantity $\hat{\mathbf{s}}^{(k+1)}$. For the GMM example, after the initialization of the quantity $\hat{\mathbf{s}}^{(0)} = n^{-1}\sum_{i=1}^{n}\overline{\mathbf{s}}_i^{(0)}$, those E-steps break down as follows:

**Batch EM (EM):** for all $i \in [\![1,n]\!]$, compute $\overline{\mathbf{s}}_i^{(k)}$ and set

$$\hat{\mathbf{s}}^{(k+1)} = n^{-1}\sum_{i=1}^{n}\overline{\mathbf{s}}_i^{(k)}. \tag{103}$$

**Online EM (sEM):** draw an index $i_k$ uniformly at random on $[\![n]\!]$, compute $\overline{\mathbf{s}}_{i_k}^{(k)}$ and set

$$\hat{\mathbf{s}}^{(k+1)} = (1-\gamma_k)\hat{\mathbf{s}}^{(k)} + \gamma_k\overline{\mathbf{s}}_{i_k}^{(k)}. \tag{104}$$

**Incremental EM (iEM):** draw an index $i_k$ uniformly at random on $[\![n]\!]$, compute $\overline{\mathbf{s}}_{i_k}^{(k)}$ and set

$$\hat{\mathbf{s}}^{(k+1)} = \hat{\mathbf{s}}^{(k)} + \overline{\mathbf{s}}_{i_k}^{(k)} - \overline{\mathbf{s}}_{i_k}^{(\tau_i^k)} = n^{-1}\sum_{i=1}^{n}\overline{\mathbf{s}}_i^{(\tau_i^k)}. \tag{105}$$

**Variance reduced stochastic EM (sEM-VR):** draw an index $i_k$ uniformly at random on $[\![n]\!]$, compute $\overline{\mathbf{s}}_{i_k}^{(k)}$ and set

$$\hat{\mathbf{s}}^{(k+1)} = (1-\gamma)\hat{\mathbf{s}}^{(k)} + \gamma\left(\overline{\mathbf{s}}_{i_k}^{(k)} - \overline{\mathbf{s}}_{i_k}^{(\ell(k))} + \overline{\mathbf{s}}^{(\ell(k))}\right) \tag{106}$$

where $\overline{\mathbf{s}}_{i_k}^{(\ell(k))}$ and $\overline{\mathbf{s}}^{(\ell(k))}$ were computed at iteration $\ell(k)$, defined as the first iteration number in the epoch that iteration $k$ is in.

**Fast Incremental EM (fiEM):** draw two different and independent indices $(i_k, j_k)$ uniformly at random on $[\![n]\!]$, compute the quantities $\overline{\mathbf{s}}_{i_k}^{(k)}$ and $\overline{\mathbf{s}}_{j_k}^{(k)}$ and set

$$\hat{\mathbf{s}}^{(k+1)} = (1-\gamma)\hat{\mathbf{s}}^{(k)} + \gamma\left(\overline{\boldsymbol{\mathcal{S}}}^{(k)} + \overline{\mathbf{s}}_{i_k}^{(k)} - \overline{\mathbf{s}}_{i_k}^{(t_{i_k}^k)}\right)$$
$$\overline{\boldsymbol{\mathcal{S}}}^{(k+1)} = \overline{\boldsymbol{\mathcal{S}}}^{(k)} + n^{-1}\left(\overline{\mathbf{s}}_{j_k}^{(k)} - \overline{\mathbf{s}}_{j_k}^{(t_{j_k}^k)}\right) \tag{107}$$

Finally, the $k$-th update reads $\hat{\boldsymbol{\theta}}^{(k+1)} = \overline{\boldsymbol{\theta}}(\hat{\mathbf{s}}^{(k+1)})$ where the function $s \to \overline{\boldsymbol{\theta}}(s)$ is defined by (96).

### G.2 Probabilistic Latent Semantic Analysis

#### G.2.1 Model assumptions

The constraint set $\Theta$ is given by

$$\Theta = \left(\times_{d\in[\![D]\!]}\Delta^K\right) \times \left(\times_{k\in[\![K]\!]}\Delta^V\right). \tag{108}$$

For the sE-step (4) in the EM methods, we compute the expected complete data statistics as

$$
\begin{aligned}
\bar{\mathbf{s}}_{i,d,k}^{(t|d)}(\boldsymbol{\theta}^{(\mathrm{t|d})},\boldsymbol{\theta}^{(\mathrm{w|t})}) &= \mathbb{1}_{\{d\}}(y_i^{(\mathrm{d})})\Big(\textstyle\sum_{\ell=1}^K \boldsymbol{\theta}_{d,\ell}^{(\mathrm{t|d})}\boldsymbol{\theta}_{\ell,y_i^{(\mathrm{w})}}^{(\mathrm{w|t})}\Big)^{-1}\boldsymbol{\theta}_{d,k}^{(\mathrm{t|d})}\boldsymbol{\theta}_{k,y_i^{(\mathrm{w})}}^{(\mathrm{w|t})}, \\
\bar{\mathbf{s}}_{i,k,v}^{(w|t)}(\boldsymbol{\theta}^{(\mathrm{t|d})},\boldsymbol{\theta}^{(\mathrm{w|t})}) &= \mathbb{1}_{\{v\}}(y_i^{(\mathrm{w})})\Big(\textstyle\sum_{\ell=1}^K \boldsymbol{\theta}_{y_i^{(\mathrm{d})},\ell}^{(\mathrm{t|d})}\boldsymbol{\theta}_{\ell,v}^{(\mathrm{w|t})}\Big)^{-1}\boldsymbol{\theta}_{y_i^{(\mathrm{d})},k}^{(\mathrm{t|d})}\boldsymbol{\theta}_{k,v}^{(\mathrm{w|t})},
\end{aligned} \tag{109}
$$

for each $(i,k,d,v) \in [\![n]\!] \times [\![K]\!] \times [\![D]\!] \times [\![V]\!]$. Meanwhile, the regularized M-step (5) in the EM methods evaluates to:

$$
\begin{pmatrix} \bar{\boldsymbol{\theta}}_{d,k}^{(\mathrm{t|d})}(\boldsymbol{s}) \\ \bar{\boldsymbol{\theta}}_{k,v}^{(\mathrm{w|t})}(\boldsymbol{s}) \end{pmatrix} = \begin{pmatrix} \left(\sum_{i=1}^n \sum_{k'=1}^K \boldsymbol{s}_{i,d,k'}^{(t|d)} + \alpha'K\right)^{-1}\left(\sum_{i=1}^n \boldsymbol{s}_{i,d,k}^{(t|d)} + \alpha'\right) \\ \left(\sum_{i=1}^n \sum_{\ell=1}^V \boldsymbol{s}_{i,k,\ell}^{(w|t)} + \beta'V\right)^{-1}\left(\sum_{i=1}^n \boldsymbol{s}_{i,k,v}^{(w|t)} + \beta'\right) \end{pmatrix}, \tag{110}
$$

for each $(k,d,v) \in [\![K]\!] \times [\![D]\!] \times [\![V]\!]$.

Using the partition of the sufficient statistics as $S(y_i,z_i) = (S^{(t|d)}(y_i,z_i)^\top, S^{(w|t)}(y_i,z_i)^\top)^\top \in \mathbb{R}^{KD+KV}$, the partition $\phi(\boldsymbol{\theta}) = (\phi^{(t|d)}(\boldsymbol{\theta})^\top, \phi^{(w|t)}(\boldsymbol{\theta})^\top)^\top \in \mathbb{R}^{KD+KV}$, the complete log-likelihood (30) can be expressed in the standard form as (2) with

$$
\begin{aligned}
\boldsymbol{s}_{i,d,k}^{(t|d)} &= \mathbb{1}_{\{k,d\}}(z_i,y_i^{(\mathrm{d})}), \quad \phi_{d,k}^{(t|d)}(\boldsymbol{\theta}) = \log(\boldsymbol{\theta}_{d,k}^{(\mathrm{t|d})}), \\
\boldsymbol{s}_{i,k,v}^{(w|t)} &= \mathbb{1}_{\{k,v\}}(z_i,y_i^{(\mathrm{w})}), \quad \phi_{k,v}^{(w|t)}(\boldsymbol{\theta}) = \log(\boldsymbol{\theta}_{k,v}^{(\mathrm{w|t})}),
\end{aligned} \tag{111}
$$

Assumption H1 is verified with $\mathsf{Z} = [\![K]\!]$ and the sufficient statistics defined in (111) that leads to a compact $\mathsf{S}$.

By using the vectorization of $\boldsymbol{\theta}$ as an $(K-1)D + (V-1)K$-dimensional vector, we can calculate the Jacobian as follows. In particular,

$$
\mathrm{J}_{\boldsymbol{\theta}_{d,k}^{(\mathrm{t|d})}}^{\phi_{d',k'}^{(t|d)}}(\boldsymbol{\theta}) = \begin{cases} 0 & \text{if } d' \neq d, \\ \frac{1}{1-\sum_{\ell=1}^{K-1}\boldsymbol{\theta}_{d,\ell}^{(\mathrm{t|d})}} & \text{if } d' = d, k' \neq k \\ \frac{1}{\boldsymbol{\theta}_{d,k}^{(\mathrm{t|d})}} & \text{if } d' = d, k' = k. \end{cases} , \quad \mathrm{J}_{\boldsymbol{\theta}_{k,v}^{(\mathrm{w|t})}}^{\phi_{k',v'}^{(w|t)}}(\boldsymbol{\theta}) = \begin{cases} 0 & \text{if } k' \neq k, \\ \frac{1}{1-\sum_{\ell=1}^{V-1}\boldsymbol{\theta}_{k,\ell}^{(\mathrm{w|t})}} & \text{if } k' = k, v' \neq v \\ \frac{1}{\boldsymbol{\theta}_{k,v}^{(\mathrm{w|t})}} & \text{if } k' = k, v' = v. \end{cases}
$$
$$\tag{112}$$

With the above definitions, it can be verified that the Jacobian matrix is full rank and smooth *w.r.t.* $\boldsymbol{\theta}$ for any $\boldsymbol{\theta} \in \mathrm{int}(\Theta)$. This confirms H2.

Next, we verify H3, *i.e.,* the Lipschitz continuity of $p(z_i|y_i;\boldsymbol{\theta})$, w.r.t to $\boldsymbol{\theta}$. Note that for all $(i,k,d) \in [\![n]\!] \times [\![K]\!] \times [\![D]\!]$, $p(z_i = k|y_i;\boldsymbol{\theta}_{d,k}^{(\mathrm{t|d})},\boldsymbol{\theta}_{k,v}^{(\mathrm{w|t})}) = \mathbb{E}_{\boldsymbol{\theta}}[\mathbb{1}_{\{k,d\}}(z_i,y_i^{(\mathrm{d})})|y_i] = \bar{\mathbf{s}}_{i,k,d}^{(t|d)}(\boldsymbol{\theta}^{(\mathrm{t|d})},\boldsymbol{\theta}^{(\mathrm{w|t})})$ as defined in (109). Observe that as we focus on $\boldsymbol{\theta} \in \mathrm{int}(\Theta)$, each of $\boldsymbol{\theta}_{d,\ell}^{(\mathrm{t|d})}\boldsymbol{\theta}_{\ell,y_i^{(\mathrm{w})}}^{(\mathrm{w|t})}$, $\boldsymbol{\theta}_{y_i^{(\mathrm{d})},\ell}^{(\mathrm{t|d})}\boldsymbol{\theta}_{\ell,v}^{(\mathrm{w|t})}$ is strictly positive and strictly less than one. The Lipschitz property follows from the expression (109).

The expression of the regularized complete log-likelihood, $\boldsymbol{\theta} \to L(s,\boldsymbol{\theta})$, is defined as:

$$
L(s,\boldsymbol{\theta}) = -\sum_{k=1}^K \sum_{d=1}^D \boldsymbol{s}_{i,k,d}^{(t|d)}\log(\boldsymbol{\theta}_{d,k}^{(\mathrm{t|d})}) - \alpha'\log(\boldsymbol{\theta}_{d,k}^{(\mathrm{t|d})}) - \sum_{k=1}^K \sum_{v=1}^V \boldsymbol{s}_{i,k,v}^{(w|t)}\log(\boldsymbol{\theta}_{k,v}^{(\mathrm{w|t})}) - \beta'\log(\boldsymbol{\theta}_{k,v}^{(\mathrm{w|t})}),
$$

This function admits a unique minimum in $\mathrm{int}(\Theta)$ from the strict concavity of the logarithm, as the regularizations are active with $\alpha', \beta' > 0$. By the same virtue of the verification of H2, we observe that H4 can be satisfied.

We first calculate the quantity $\mathrm{B}(\mathbf{s})$ defined in (17). Using the vectorization of $\boldsymbol{\theta}$ as a $(K-1)D + (V-1)K$-dimensional vector, we observe that the Hessian of the function $\boldsymbol{\theta} \mapsto L(\mathbf{s}, \boldsymbol{\theta})$ w.r.t. to $\boldsymbol{\theta}$ has a block diagonal structure with $D+K$ blocks — the $d$th block which corresponds to $\boldsymbol{\theta}_{d,\cdot}^{(\mathrm{t|d})}$ is given by

$$\left[ \mathrm{H}_L^{\boldsymbol{\theta}}(\mathbf{s}, \boldsymbol{\theta}) \right]_d = \frac{\boldsymbol{s}_{K,d}^{(t|d)} + \alpha'}{(1 - \sum_{k=1}^{K-1} \boldsymbol{\theta}_{d,k}^{(\mathrm{t|d})})^2} \mathbf{1}\mathbf{1}^\top + \mathrm{Diag}(\frac{\boldsymbol{s}^{(t|d)} + \alpha'\mathbf{1}}{(\boldsymbol{\theta}^{(\mathrm{t|d})})^2}) \tag{113}$$

while the $(D+k)$th block which corresponds to $\boldsymbol{\theta}_{k,\cdot}^{(\mathrm{w|t})}$ is given by

$$\left[ \mathrm{H}_L^{\boldsymbol{\theta}}(\mathbf{s}, \boldsymbol{\theta}) \right]_{D+k} = \frac{\boldsymbol{s}_{k,V}^{(w|t)} + \beta'}{(1 - \sum_{\ell=1}^{V-1} \boldsymbol{\theta}_{k,\ell}^{(\mathrm{w|t})})^2} \mathbf{1}\mathbf{1}^\top + \mathrm{Diag}(\frac{\boldsymbol{s}^{(w|t)} + \beta'\mathbf{1}}{(\boldsymbol{\theta}^{(\mathrm{w|t})})^2}) \tag{114}$$

Since each block in the above Hessian matrix is positive definite, the matrix

$$\mathrm{B}(\mathbf{s}) := \mathrm{J}_\phi^{\boldsymbol{\theta}}(\overline{\boldsymbol{\theta}}(\mathbf{s})) \left( \mathrm{H}_L^{\boldsymbol{\theta}}(\mathbf{s}, \overline{\boldsymbol{\theta}}(\mathbf{s})) \right)^{-1} \mathrm{J}_\phi^{\boldsymbol{\theta}}(\overline{\boldsymbol{\theta}}(\mathbf{s}))^\top = \boldsymbol{\mathcal{J}}(s)\boldsymbol{\mathcal{J}}(s)^\top \tag{115}$$

is positive definite and bounded. Furthermore, there exists a constant $\mathrm{L}_B$ such that $\| \mathrm{B}(\mathbf{s}) - \mathrm{B}(\mathbf{s}') \| \leq \mathrm{L}_B \|\mathbf{s} - \mathbf{s}'\|$. Finally, this confirms H5.

### G.2.2 Algorithms updates

In the sequel, for all $(i, d, k, v) \in [\![n]\!] \times [\![D]\!] \times [\![K]\!] \times [\![V]\!]$ the conditional expectations $\overline{\mathbf{s}}_{i,k,d}^{(t|d)}((\boldsymbol{\theta}_{d,k}^{(\mathrm{t|d})})^\delta, (\boldsymbol{\theta}_{k,v}^{(\mathrm{w|t})})^\delta)$ and $\overline{\mathbf{s}}_{i,k,v}^{(w|t)}((\boldsymbol{\theta}_{d,k}^{(\mathrm{t|d})})^\delta, (\boldsymbol{\theta}_{k,v}^{(\mathrm{w|t})})^\delta)$ are defined by (109). For the pLSA example, after the initialization of the quantity $(s_{k,d}^{(1)})^0 = n^{-1} \sum_{i=1}^n \overline{\mathbf{s}}_{i,k,d}^{(t|d)}((\boldsymbol{\theta}_{d,k}^{(\mathrm{t|d})})^0, (\boldsymbol{\theta}_{k,v}^{(\mathrm{w|t})})^0)$ and $(s_{k,v}^{(2)})^0 = n^{-1} \sum_{i=1}^n \overline{\mathbf{s}}_{i,k,v}^{(w|t)}((\boldsymbol{\theta}_{d,k}^{(\mathrm{t|d})})^0, (\boldsymbol{\theta}_{k,v}^{(\mathrm{w|t})})^0)$, the several E-steps break down as follows:

**Batch EM (EM):** At iteration $\delta$: update the statistics for all $(d, k, v) \in [\![D]\!] \times [\![K]\!] \times [\![V]\!]$ :

$$(\boldsymbol{s}_{k,d}^{(1)})^{\delta+1} = \sum_{i=1}^n \overline{\mathbf{s}}_{i,k,d}^{(t|d)}((\boldsymbol{\theta}_{d,k}^{(\mathrm{t|d})})^\delta, (\boldsymbol{\theta}_{k,v}^{(\mathrm{w|t})})^\delta) \quad \text{and} \quad (\boldsymbol{s}_{k,v}^{(2)})^{\delta+1} = \sum_{i=1}^n \overline{\mathbf{s}}_{i,k,v}^{(w|t)}((\boldsymbol{\theta}_{d,k}^{(\mathrm{t|d})})^\delta, (\boldsymbol{\theta}_{k,v}^{(\mathrm{w|t})})^\delta) \tag{116}$$

**Online EM (sEM):** At iteration $\delta$, update the statistics for all $(d, k, v) \in [\![D]\!] \times [\![K]\!] \times [\![V]\!]$ :

$$\begin{aligned} (\boldsymbol{s}_{k,d}^{(1)})^{\delta+1} &= (1 - \gamma_\delta)(\boldsymbol{s}_{k,d}^{(1)})^\delta + \gamma_\delta \overline{\mathbf{s}}_{i_\delta,k,d}^{(t|d)}((\boldsymbol{\theta}_{d,k}^{(\mathrm{t|d})})^\delta, (\boldsymbol{\theta}_{k,v}^{(\mathrm{w|t})})^\delta) \\ (\boldsymbol{s}_{k,v}^{(2)})^{\delta+1} &= (1 - \gamma_\delta)(\boldsymbol{s}_{k,v}^{(2)})^\delta + \gamma_\delta \overline{\mathbf{s}}_{i_\delta,k,v}^{(w|t)}((\boldsymbol{\theta}_{d,k}^{(\mathrm{t|d})})^\delta, (\boldsymbol{\theta}_{k,v}^{(\mathrm{w|t})})^\delta) \end{aligned} \tag{117}$$

**Incremental EM (iEM):** At iteration $\delta$, update the statistics for all $(d, k, v) \in [\![D]\!] \times [\![K]\!] \times [\![V]\!]$ :

$$\begin{aligned} (\boldsymbol{s}_{k,d}^{(1)})^{\delta+1} &= (\boldsymbol{s}_{k,d}^{(1)})^\delta + \overline{\mathbf{s}}_{i_\delta,k,d}^{(t|d)}((\boldsymbol{\theta}_{d,k}^{(\mathrm{t|d})})^\delta, (\boldsymbol{\theta}_{k,v}^{(\mathrm{w|t})})^\delta) - \overline{\mathbf{s}}_{i_\delta,k,d}^{(t|d)}((\boldsymbol{\theta}_{d,k}^{(\mathrm{t|d})})^{\tau_{i_\delta}^\delta}, (\boldsymbol{\theta}_{k,v}^{(\mathrm{w|t})})^{\tau_{i_\delta}^\delta}) \\ (\boldsymbol{s}_{k,v}^{(2)})^{\delta+1} &= (\boldsymbol{s}_{k,v}^{(2)})^\delta + \overline{\mathbf{s}}_{i_\delta,k,v}^{(w|t)}((\boldsymbol{\theta}_{d,k}^{(\mathrm{t|d})})^\delta, (\boldsymbol{\theta}_{k,v}^{(\mathrm{w|t})})^\delta) - \overline{\mathbf{s}}_{i_\delta,k,v}^{(w|t)}((\boldsymbol{\theta}_{d,k}^{(\mathrm{t|d})})^{\tau_{i_\delta}^\delta}, (\boldsymbol{\theta}_{k,v}^{(\mathrm{w|t})})^{\tau_{i_\delta}^\delta}) \end{aligned} \tag{118}$$

**Variance reduced stochastic EM (sEM-VR):** At iteration $\delta$, draw an index $i_\delta$ and update the statistics for all $(d, k, v) \in [\![D]\!] \times [\![K]\!] \times [\![V]\!]$ :

$$(\boldsymbol{s}_{k,d}^{(1)})^{\delta+1} = (1 - \gamma)(\boldsymbol{s}_{k,d}^{(1)})^\delta$$
$$+ \gamma \Big( \overline{\mathbf{s}}_{i_\delta,k,d}^{(t|d)}((\boldsymbol{\theta}_{d,k}^{(\mathrm{t|d})})^\delta, (\boldsymbol{\theta}_{k,v}^{(\mathrm{w|t})})^\delta) - \overline{\mathbf{s}}_{i_\delta,k,d}^{(t|d)}((\boldsymbol{\theta}_{d,k}^{(\mathrm{t|d})})^{(\ell(k))}, (\boldsymbol{\theta}_{k,v}^{(\mathrm{w|t})})^{(\ell(k))}) + \overline{\mathbf{s}}^{(t|d)}((\boldsymbol{\theta}_{d,k}^{(\mathrm{t|d})})^{(\ell(k))}, (\boldsymbol{\theta}_{k,v}^{(\mathrm{w|t})})^{(\ell(k))}) \Big)$$
$$(\boldsymbol{s}_{k,v}^{(2)})^{\delta+1} = (1 - \gamma)(\boldsymbol{s}_{k,v}^{(2)})^\delta$$
$$+ \gamma \Big( \overline{\mathbf{s}}_{i_\delta,k,v}^{(w|t)}((\boldsymbol{\theta}_{d,k}^{(\mathrm{t|d})})^\delta, (\boldsymbol{\theta}_{k,v}^{(\mathrm{w|t})})^\delta) - \overline{\mathbf{s}}_{i_\delta,k,v}^{(w|t)}((\boldsymbol{\theta}_{d,k}^{(\mathrm{t|d})})^{(\ell(k))}, (\boldsymbol{\theta}_{k,v}^{(\mathrm{w|t})})^{(\ell(k))}) + \overline{\mathbf{s}}^{(w|t)}((\boldsymbol{\theta}_{d,k}^{(\mathrm{t|d})})^{(\ell(k))}, (\boldsymbol{\theta}_{k,v}^{(\mathrm{w|t})})^{(\ell(k))}) \Big)$$

$$\tag{119}$$

**Fast Incremental EM (fiEM):** At iteration $\delta$, draw two indices $(i_\delta, j_\delta)$ independently and update the statistics for all $(d, k, v) \in [\![D]\!] \times [\![K]\!] \times [\![V]\!]$ :

$$\left(s_{k,d}^{(1)}\right)^{\delta+1} = (1-\gamma)\left(s_{k,d}^{(1)}\right)^{\delta}$$
$$+ \gamma\left(\overline{\mathbf{s}}_{i_\delta,k,d}^{(\mathrm{t}|d)}\big((\boldsymbol{\theta}_{d,k}^{(\mathrm{t}|\mathrm{d})})^{\delta}, (\boldsymbol{\theta}_{k,v}^{(\mathrm{w}|\mathrm{t})})^{\delta}\big) - \overline{\mathbf{s}}_{i_\delta,k,d}^{(\mathrm{t}|d)}\big((\boldsymbol{\theta}_{d,k}^{(\mathrm{t}|\mathrm{d})})^{(t_{i_\delta}^{\delta})}, (\boldsymbol{\theta}_{k,v}^{(\mathrm{w}|\mathrm{t})})^{(t_{i_\delta}^{\delta})}\big) + \left(\overline{\boldsymbol{S}}_{k,d}^{(1)}\right)^{\delta}\right)$$

$$\left(\overline{\boldsymbol{S}}_{k,d}^{(1)}\right)^{\delta+1} = \left(\overline{\boldsymbol{S}}_{k,d}^{(1)}\right)^{\delta} + n^{-1}\left(\overline{\mathbf{s}}_{j_\delta,k,d}^{(\mathrm{t}|d)}\big((\boldsymbol{\theta}_{d,k}^{(\mathrm{t}|\mathrm{d})})^{\delta}, (\boldsymbol{\theta}_{k,v}^{(\mathrm{w}|\mathrm{t})})^{\delta}\big) - \overline{\mathbf{s}}_{j_\delta,k,d}^{(\mathrm{t}|d)}\big((\boldsymbol{\theta}_{d,k}^{(\mathrm{t}|\mathrm{d})})^{(t_{j_\delta}^{\delta})}, (\boldsymbol{\theta}_{k,v}^{(\mathrm{w}|\mathrm{t})})^{(t_{j_\delta}^{\delta})}\big)\right)$$

$$\left(s_{k,v}^{(2)}\right)^{\delta+1} = (1-\gamma)\left(s_{k,v}^{(2)}\right)^{\delta}$$
$$+ \gamma\left(\overline{\mathbf{s}}_{i_\delta,k,v}^{(\mathrm{t}|d)}\big((\boldsymbol{\theta}_{d,k}^{(\mathrm{t}|\mathrm{d})})^{\delta}, (\boldsymbol{\theta}_{k,v}^{(\mathrm{w}|\mathrm{t})})^{\delta}\big) - \overline{\mathbf{s}}_{i_\delta,k,v}^{(\mathrm{t}|d)}\big((\boldsymbol{\theta}_{d,k}^{(\mathrm{t}|\mathrm{d})})^{(t_{i_\delta}^{\delta})}, (\boldsymbol{\theta}_{k,v}^{(\mathrm{w}|\mathrm{t})})^{(t_{i_\delta}^{\delta})}\big) + \left(\overline{\boldsymbol{S}}_{k,v}^{(2)}\right)^{\delta}\right)$$

$$\left(\overline{\boldsymbol{S}}_{k,v}^{(2)}\right)^{\delta+1} = \left(\overline{\boldsymbol{S}}_{k,v}^{(2)}\right)^{\delta} + \gamma n^{-1}\left(\overline{\mathbf{s}}_{j_\delta,k,v}^{(\mathrm{t}|d)}\big((\boldsymbol{\theta}_{d,k}^{(\mathrm{t}|\mathrm{d})})^{\delta}, (\boldsymbol{\theta}_{k,v}^{(\mathrm{w}|\mathrm{t})})^{\delta}\big) - \overline{\mathbf{s}}_{j_\delta,k,v}^{(\mathrm{t}|d)}\big((\boldsymbol{\theta}_{d,k}^{(\mathrm{t}|\mathrm{d})})^{(t_{j_\delta}^{\delta})}, (\boldsymbol{\theta}_{k,v}^{(\mathrm{w}|\mathrm{t})})^{(t_{j_\delta}^{\delta})}\big)\right)$$

Finally, at iteration $\delta$, for $(k, d, v) \in [\![K]\!] \times [\![D]\!] \times [\![V]\!]$, the M-step in (5) evaluates to:

$$\begin{pmatrix} (\boldsymbol{\theta}_{d,k}^{(\mathrm{t}|\mathrm{d})})^{\delta+1} \\ (\boldsymbol{\theta}_{k,v}^{(\mathrm{w}|\mathrm{t})})^{\delta+1} \end{pmatrix} = \begin{pmatrix} \big(\sum_{k'=1}^{K} (s_{k',d}^{(1)})^{\delta+1} + \alpha'K\big)^{-1}\big((s_{k,d}^{(1)})^{\delta+1} + \alpha'\big) \\ \big(\sum_{\ell=1}^{V} (s_{k,\ell}^{(2)})^{\delta+1} + \beta'V\big)^{-1}\big((s_{k,v}^{(2)})^{\delta+1} + \beta'\big) \end{pmatrix}. \tag{120}$$

# H  Local Linear Convergence of fiEM

In this section, we prove that the fiEM method converges locally at a linear rate to a stationary point, under a similar set of assumptions as in [Chen et al., 2018]. Note that some of the following assumptions can be difficult to verify, and our analysis here is merely a proof of concept.

Consider a stationary point $\boldsymbol{\theta}^\star$ to problem (1) and its corresponding sufficient statistics $\mathbf{s}^\star$, also a stationary point to (22). To simplify notations, we follow [Chen et al., 2018] and write the complete sufficient statistics as $F(\mathbf{s}') := \overline{\mathbf{s}}(\overline{\boldsymbol{\theta}}(\mathbf{s}'))$, and also the $i$th sufficient statistics as $f_i(\mathbf{s}') := \overline{\mathbf{s}}_i(\overline{\boldsymbol{\theta}}(\mathbf{s}'))$. We assume the following:

**B1.** *The Hessian matrix $\nabla^2 \overline{\mathcal{L}}(\boldsymbol{\theta}^\star)$ is strictly positive definite such that $\boldsymbol{\theta}^\star$ is a strict local minimizer of problem (1).*

**B2.** *For any $k \geq 1$, we have $\|\hat{\mathbf{s}}^k - \mathbf{s}^\star\| \leq \frac{\lambda}{\mathrm{L}_\mathbf{s}}$, where $\mathrm{L}_\mathbf{s}$ was defined in our Lemma 4 and $1 - \lambda$ is the maximum eigenvalue of the Jacobian matrix $\mathrm{J}_F^\mathbf{s}(\mathbf{s}^\star)$.*

The above assumptions correspond to assumptions (a), (c) in [Chen et al., 2018, Theorem 1], while we note that assumption (b) therein are shown in our Lemma 4.

We remark that B1 is strictly stronger than H4 used in our global convergence analysis. The latter makes assumption on the actual objective function $\overline{\mathcal{L}}(\boldsymbol{\theta}^\star)$ instead of the surrogate function $\boldsymbol{\theta} \to L(\mathbf{s}, \boldsymbol{\theta})$. Our proof goes as follows.

**Proposition 1.** *Under Assumption B1, B2 and the conditions such that our Lemma 4 holds. The fiEM method converges linearly such that*

$$\mathbb{E}[\|\hat{\boldsymbol{s}}^{(k+1)} - \mathbf{s}^\star\|^2] \leq (1-\delta)^{k+1}\|\hat{\boldsymbol{s}}^{(0)} - \mathbf{s}^\star\|^2, \ \forall \ k \geq 0, \tag{121}$$

*where $\delta = \Theta(1/n)$ with an appropriately chosen step size $\gamma$.*

**Proof** (Sketch) For $k \in \mathbb{N}^*$, denote by $\mathcal{F}_k$ the $\sigma$-algebra generated by the random variables $i_0, j_0, \ldots, i_k, j_k$. Consider

$$\mathbb{E}\big[\|\hat{\boldsymbol{s}}^{(k+1)} - \mathbf{s}^\star\|^2|\mathcal{F}_k\big] = \mathbb{E}\big[\|\hat{\boldsymbol{s}}^{(k)} - \gamma(\hat{\boldsymbol{s}}^{(k)} - \boldsymbol{S}^{(k+1)}) - \mathbf{s}^\star\|^2|\mathcal{F}_k\big]$$
$$= \mathbb{E}\big[\|(1-\gamma)\hat{\boldsymbol{s}}^{(k)} + \gamma F(\hat{\boldsymbol{s}}^{(k)}) - \mathbf{s}^\star + \gamma\big(\boldsymbol{S}^{(k+1)} - F(\hat{\boldsymbol{s}}^{(k)})\big)\|^2|\mathcal{F}_k\big] \tag{122}$$

Note that as $\mathbb{E}[\boldsymbol{S}^{(k+1)} - F(\hat{\boldsymbol{s}}^{(k)})|\mathcal{F}_k] = 0$, we have

$$\mathbb{E}\big[\|\hat{\boldsymbol{s}}^{(k+1)} - \mathbf{s}^\star\|^2|\mathcal{F}_k\big]$$
$$= \mathbb{E}\big[\|(1-\gamma)\hat{\boldsymbol{s}}^{(k)} + \gamma F(\hat{\boldsymbol{s}}^{(k)}) - \mathbf{s}^\star\|^2|\mathcal{F}_k\big] + \gamma^2 \mathbb{E}\big[\|\boldsymbol{S}^{(k+1)} - F(\hat{\boldsymbol{s}}^{(k)})\|^2|\mathcal{F}_k\big] \tag{123}$$

Repeating the analysis in (9) of [Chen et al., 2018], we arrive at the upper bound

$$\mathbb{E}\big[\|(1-\gamma)\hat{\boldsymbol{s}}^{(k)} + \gamma F(\hat{\boldsymbol{s}}^{(k)}) - \mathbf{s}^\star\|^2 | \mathcal{F}_k\big] \le (1-\gamma\lambda/2)\|\hat{\boldsymbol{s}}^{(k)} - \mathbf{s}^\star\|^2 \tag{124}$$

On the other hand, applying [Defazio et al., 2014, Lemma 3] shows that

$$
\begin{aligned}
\mathbb{E}\big[\|\boldsymbol{\mathcal{S}}^{(k+1)} - F(\hat{\boldsymbol{s}}^{(k)})\|^2 | \mathcal{F}_k\big] &\le 2\Big(\|f_{i_k}(\hat{\boldsymbol{s}}^{(\tau_{i_k}^k)}) - f_{i_k}(\mathbf{s}^\star)\|^2 + \|f_{i_k}(\hat{\boldsymbol{s}}^{(k)}) - f_{i_k}(\mathbf{s}^\star)\|^2\Big)\\
&\le 2\,\mathrm{L}_{\mathbf{s}}^2\left(\|\hat{\boldsymbol{s}}^{(\tau_{i_k}^k)} - \mathbf{s}^\star\|^2 + \|\hat{\boldsymbol{s}}^{(k)} - \mathbf{s}^\star\|^2\right)
\end{aligned}
\tag{125}
$$

Denote the total expectation as $h_k := \mathbb{E}[\|\hat{\boldsymbol{s}}^{(k)} - \mathbf{s}^\star\|^2]$, and taking the total expectation on both sides yields

$$\mathbb{E}\big[\|\boldsymbol{\mathcal{S}}^{(k+1)} - F(\hat{\boldsymbol{s}}^{(k)})\|^2\big] \le 2\,\mathrm{L}_{\mathbf{s}}^2\left(h_k + \tfrac{1}{n}\sum_{i=1}^n h_{\tau_i^k}\right) \tag{126}$$

Substituting the above into (123) yields

$$h_{k+1} \le \left(1 - \gamma\frac{\lambda}{2} + 2\gamma^2\,\mathrm{L}_{\mathbf{s}}^2\right)h_k + 2\gamma^2\,\mathrm{L}_{\mathbf{s}}^2\left(\tfrac{1}{n}\sum_{i=1}^n h_{\tau_i^k}\right) \tag{127}$$

Moreover, we observe the following recursion through evaluating the expectation

$$\frac{1}{n}\sum_{i=1}^n h_{\tau_i^k} = \frac{1}{n}h_{k-1} + \left(1-\frac{1}{n}\right)\frac{1}{n}\sum_{i=1}^n h_{\tau_i^{k-1}} \le \frac{1}{n}\sum_{\ell=0}^{k-1}\left(1-\frac{1}{n}\right)^{k-\ell-1} h_\ell \tag{128}$$

Therefore, (127) simplifies to

$$h_{k+1} \le \left(1-\gamma\frac{\lambda}{2}+2\gamma^2\,\mathrm{L}_{\mathbf{s}}^2\right)h_k + \frac{2\gamma^2\,\mathrm{L}_{\mathbf{s}}^2}{n}\sum_{\ell=0}^{k-1}\left(1-\frac{1}{n}\right)^{k-\ell-1} h_\ell \tag{129}$$

To this end, we let $a = \frac{\lambda}{2}, b = 2\,\mathrm{L}_{\mathbf{s}}^2, c = 2\,\mathrm{L}_{\mathbf{s}}^2$ and consider the following inequality,

$$h_{k+1} \le \left(1-\gamma a+\gamma^2 b\right)h_k + \frac{\gamma^2 c}{n}\sum_{\ell=0}^{k-1}\left(1-\frac{1}{n}\right)^{k-\ell-1} h_\ell \tag{130}$$

We claim that for a sufficiently small step size $\gamma$, there exists $\delta \in (0,1]$ such that $h_k \le (1-\delta)^k h_0$ for all $k$. The proof can be achieved using induction. The base case is straightforward since:

$$h_1 \le (1-\gamma a+\gamma^2 b)h_0 \tag{131}$$

For the induction case, we assume that $h_\tau \le (1-\delta)^\tau h_0$ for $\tau = 1, 2, ..., k$. We observe that the induction hypothesis implies

$$
\begin{aligned}
\frac{h_{k+1}}{h_0} &\le \left(1-\gamma a+\gamma^2 b\right)(1-\delta)^k + \frac{\gamma^2 c}{n}\sum_{\ell=0}^{k-1}\left(1-\frac{1}{n}\right)^{k-\ell-1}(1-\delta)^\ell\\
&\le \left(1-\gamma a+\gamma^2 b\right)(1-\delta)^k + \frac{\gamma^2 c}{n}(1-\delta)^{k-1}\frac{1}{1-\frac{1-1/n}{1-\delta}}\\
&= (1-\delta)^k\left\{\left(1-\gamma a+\gamma^2 b\right)+\gamma^2 c\frac{1}{1-n\delta}\right\}\\
&\stackrel{(a)}{\approx} (1-\delta)^k\left\{\left(1-\gamma a+\gamma^2 b\right)+\gamma^2 c(1+\delta n)\right\}\\
&\le (1-\delta)^k\left\{\left(1-\gamma a+\gamma^2 b\right)+\gamma^2 c(1+n)\right\}
\end{aligned}
\tag{132}
$$

where the approximation holds if $n\delta \ll 1$. Lastly, if

$$\gamma \le \frac{a}{2}(b+c(1+n))^{-1} \tag{133}$$

Then $h_{k+1} \le (1-\delta)^{k+1}h_0$ with $\delta \le \gamma a - \gamma^2(b+c(1+n)) = \mathcal{O}(1/n)$. □