[Reviews · NeurIPS 2019]

Reviewer 1



This paper has provided global convergence analyses, with convergence rates, of stochastic EM-algorithms which include incremental (iEM) and variance reduced (sEM-VR) versions of EM-algorithms. Especially, the paper has given a convergence rate of $O(n/\epsilon)$ for iEM by applying the theory developed by Miral(2015) and a convergence rate of $O(n^{2/3}/\epsilon)$ for sEM-VR by showing sEM-VR is a special case of stochastic scaled-gradient methods. In addition, a new variance reduced EM-algorithm named fiEM based on SAGA has been proposed with its convergence analysis as well as sEM-VR. Finally, the superiority of variance reduced variants (sEM-VR and fiEM) has been shown via numerical experiments. Clarity: The paper is clear and well written. Quality: The work is of good quality and is technically sound. Originality: Although proof techniques are not so novel from the optimization literature, this type of the convergence result of stochastic EM-algorithm is first in this context except for a few studies. The following paper is missing, which also provided non-asymptotic convergence analysis of stochastic EM-algorithms as a special case of their results. I want the author to discuss the relationship with this study. - B. Karimi, B. Miasojedow, E. Moulines, and H. Wai. Non-asymptotic Analysis of Biased Stochastic Approximation Scheme. COLT, 2019. Significance: A convergence rate of iEM is not surprising; indeed, it corresponds to that of vanilla gradient descent. However, a convergence rate of variance reduced variants is interesting because the dependency on $n$ (the number of examples) is much improved by the variance reduction. Thus, I think this paper makes a certain technical contribution to the machine learning community. A minor concern is a mismatch between this convergence rate and empirical convergence rate. Concretely, an empirical convergence rate of sEM-VR and fiEM seems linear as shown in Fig. 1. ----- I have read the authors' response which well addressed my question, so I raise the score to 7

Reviewer 2



This paper focuses on incremental EM methods for large datasets. The authors discuss the relationships between some previous incremental EM methods and propose a new incremental EM algorithm, the fiEM algorithm. The authors analyze the non-asymptotic global convergence rates of these algorithms, and compare them empirically. Originality: 1. The new algorithm fiEM is an interesting extension of the original algorithm iEM [1]. It is a good extension since it is not complicated but efficient. Quality: 1. The global convergence non-asymptotic rate analysis for the incremental EM algorithms is interesting. It showed us that the sEM-VR algorithm [3] and the new algorithm fiEM are faster than the previous algorithm iEM [1] (in expectation), which is an interesting result. It would be more interesting if the authors can provide a more detailed comparison between the sEM-VR algorithm and the fiEM algorithm since they all require O(n^{2/3}/\epsilon) number of iterations (in expectation). 2. Most of the assumptions that the authors make for the proof are reasonable, except for that I am not sure if H1 is satisfied in many settings. The sets S and Z may not be compact in many settings. 3. Empirically, the fiEM algorithm and the sEM-VR algorithm work well compared to other methods. However, on one of the real datasets in Section 4.2, the fiEM algorithm was outperformed by the previous algorithm sEM-VR. Clarity: 1. The notation of this paper is clear, but not very easy to follow since the authors used some notations before definition. It would be better if the authors can adjust the text for easier understanding. Significance: 1. The global convergence rate analysis is interesting and useful, although the experiment results may need to be improved. Also, it will be better if the authors can work on more models and datasets on the experiments. References: [1] Neal, Radford M., and Geoffrey E. Hinton. "A view of the EM algorithm that justifies incremental, sparse, and other variants." Learning in graphical models. Springer, Dordrecht, 1998. 355-368. [2] Mairal, Julien. "Incremental majorization-minimization optimization with application to large-scale machine learning." SIAM Journal on Optimization 25.2 (2015): 829-855. [3] Chen, Jianfei, et al. "Stochastic Expectation Maximization with variance reduction." Advances in Neural Information Processing Systems. 2018.

Reviewer 3



Summary: This paper studies the convergence of incremental and stochastic Expectation-Maximization (EM) algorithms. Authors establish non-asymptotic bound for the averaged EM iterates by building on the framework developed by Mairal 2015. Authors use the term 'global convergence' yet the rates are given for a first-order critical point and assuming that there is a unique minimum. ** Major comments: - Under H4, the problem already has a unique global minimum with positive definite Hessian. Any first-order critical point will be a global minimum. The result is only global because of this assumption. - Authors use the MISO framework; yet no description of MISO is provided in the paper. - In the theorems, it is not clear what the expectations are over. For example in E[\bar{L}(\hat\theta_0)], which random variable is this expectation over? - The bound is given for E[\nabla L (\theta^K)] where K is a uniform random variable between 0 and the last iteration K_max. This means that the bound is simply on 1/K_max \sum_k E[\nabla L (\theta^k)]. Using their theorems authors can give an upper bound on the averaged iterates or by using the inequality 1/K_max \sum_k E[\nabla L (\theta^k)] \geq min_k E[\nabla L (\theta^k)] the best iterate. The current way of writing the bound seems a bit opaque. At least a discussion or a brief remark on the implications of the result is needed. - Theorem 1 of Paper 1610 is a special case of Theorem 1 in Paper 1613. ** Minor comments: - line 1, algorithm -> algorithms - line 20, comma after g - line 65, to find to - line 156, tailored %%%%%%%%%%%% I thank the authors for answering my questions. Authors clarified my concern on assumption H4. But my concern on Theorem 1 of submission 1613 still remains. Based on this I am updating my score to 5.

[Author Response · NeurIPS 2019]

We would like to thank three reviewers for their feedback. Upon acceptance, we will include in the final version (a)
*new local linear convergence results for fiEM method*, (b) *an improved presentation of main results* and (c) *missing*
*references*. We first discuss a few common concerns shared by **reviewer 1**, **reviewer 2**, **reviewer 3**.

• • **Local Linear Convergence of fiEM**: As observed by the reviewers, empirically fiEM shows a local linear
convergence similar to sEM-VR. We found that fiEM has local linear convergence **in theory**, too. The new analysis
requires same assumptions as [Thm.1,Chen+2018] and adopts proof of [Defazio+2014]. We show $\mathbb{E}\|\hat{\mathbf{s}}^{(k)} - \mathbf{s}^\star\|^2 \leq$
$(1-\delta)^k\|\hat{\mathbf{s}}^{(0)} - \mathbf{s}^\star\|^2$ for $k \geq 0$ with $\delta = \Theta(1/n)$, where $\mathbf{s}^\star$ is a stationary point to (19).

• • **Satisfaction of Assumptions**: *All* assumptions H1-H5 are verified rigorously in the GMM, pLSA applications
presented, as proven in Appendix G. They are mild even though should be checked on a case-by-case basis. Reviewers
are referred to [McLachlan&Krishnan 2007] which shows satisfaction of similar assumptions on a variety of applications.

• • **Clarity:** We admit it is a challenging task to present all technical results within the page limit, but we will try our
best to improve in the final version, viz. using a running example to illustrate the assumptions used and implementation
of algorithms. We will also clarify about the expectation operators in theorems and correct typos.

**Reviewer 1:** We thank the reviewer for valuable comments and references. Our point-to-point response is as follows:

**Related work:** The paper [Karimi+2019] is relevant and will be included. Thank you for bringing it to our attention.
Karimi+[2019] focused on a biased stochastic approximation scheme and gave a global convergence rate for sEM. In
this case, their analysis shares similar scaled gradient interpretation as fiEM and sEM-VR, yet w/o variance reduction.

**iEM's Rate**: You are right as the rate of iEM is comparable to GD. Yet, iEM is a popular method without a previously
known global rate. Indeed, the comparison of iEM to fiEM, sEM-VR (theoretical & empirical) is our main contribution.

**Comparison to [Chen+2018]**: Our assumptions are more practical and less restrictive. Global convergence to stationary
point for sEM-VR in [Thm.2, Chen+2018] assumes i) the sufficient statistics $\mathbf{s}_i(\overline{\boldsymbol{\theta}}(\boldsymbol{s}'))$ is $L_s$-Lipschitz continuous in $\boldsymbol{s}'$,
$\forall i$ – this is implied by our H1-H5 via Lemma 4; ii) the complete log-likelihood is strongly concave – this is slightly
relaxed in our H4 which only requires a unique global minimizer for the complete log-likelihood. Besides, H1-H5 are
**directly verifiable** (as explained above) and we provide the rate towards a stationary point. Lastly, local convergence in
[Thm.1, Chen+2018] requires $\|\hat{\mathbf{s}}^{(k)} - \mathbf{s}^\star\|$ to be in a ball of radius $\mathcal{O}(1/L_s)$ for **any** $k \geq 1$. This is a strong assumption
that is not directly verifiable even if $\hat{\mathbf{s}}^{(0)} \approx \mathbf{s}^\star$ is known a-priori.

**Reviewer 2:** We thank the reviewer for useful comments. Please find the comparison of fiEM, sEM-VR below:

**Comparing fiEM to sEM-VR:** This comparison is analogous to comparing SAGA to SVRG for finite sum optimization,
and there is no clear ordering. In short, it depends on the trade-off of memory imprint and computation complexity.
sEM-VR requires $\mathcal{O}(\dim(\mathsf{S}))$ space to store $\overline{\mathbf{s}}^{(\ell(k))}$, yet a *full pass* on the data set is needed at each epoch, resulting in
higher complexity; meanwhile, fiEM only processes the data set *incrementally*, but it requires $\mathcal{O}(n\dim(\mathsf{S}))$ to store the
variables involved. We remark that the global rate for sEM-VR is **not proven** in [Chen+2018]. In Fig. 2 we show fiEM
outperformed sEM-VR in one dataset (a bigger one) but was outperformed by sEM-VR in the other (a smaller one).

**Reviewer 3:** We thank the reviewer for the comments. We clarify that *in addition* to analyzing iEM using the MISO
framework (which will be mentioned explicitly), we analyzed fiEM, sEM-VR with a **completely different** framework
w/ **scaled gradient**, the latter constitutes our main contribution of fast global convergence rates; see p.2 of our paper.

**Global Convergence & H4:** We emphasize H4 **does not** imply that every stationary point of (1) is global minima,
as having a unique global minimizer *does not* imply *any first order critical point is global minima*. Also, H4 refers
to *complete log-likelihood* $L(\boldsymbol{s}, \boldsymbol{\theta})$ with fixed $\boldsymbol{s}$, instead of incomplete log-likelihood $\overline{\mathcal{L}}(\boldsymbol{\theta})$ in (1). It holds for most
exponential family models where EM is useful [McLachlan&Krishnan 2007]. Mind that $\overline{\mathcal{L}}(\boldsymbol{\theta})$ is non-convex and our
convergence is *global* in the sense that it does not restrict the initialization, a common assumption for analysis of EM.

**Bounds in theorems:** The current presentation style of theorems, which evaluates the gradient norm of a ran-
domly terminated stochastic EM solution, is common in **stochastic non-convex** optimization e.g., [Ghadimi&Lan
2013,Reddi+2016a/b]. Part of the reason is that it results in a practical solution. While picking the best iterate leads to
the same sublinear rate as ours, doing so involves a full pass on the data ($\nabla\overline{\mathcal{L}}$) and computing the incomplete likelihood,
both are **difficult** tasks avoided in stochastic EM methods. Besides, as the reviewer mentioned, both random termination
and best iterate schemes lead to a quantity upper bounded by $\sum_k \mathbb{E}\|\nabla\overline{\mathcal{L}}(\boldsymbol{\theta}^{(k)})\|^2/K_{\max}$. This quantity is not equal to
the *averaged iterate*, and upper bounding it by $\mathcal{O}(1/K_{\max})$ is a non-trivial task – and is precisely our main contribution.

**Theorem 1 of Paper 1613:** Indeed, Theorem 1 for iEM is a special case of [Thm. 1, 1613]. We will cite the latter
properly. Our main contribution here lies on *fast convergence* of fiEM, sEM-VR shown by a different framework, see
Theorem 2. Detailed comments about the difference between this paper and 1613 has been sent to the AC.

[Meta-Review · NeurIPS 2019]

This paper studies the convergence of incremental and stochastic Expectation-Maximization (EM). This paper is on the borderline and was carefully discussed. The main concern of the reviewers is that Theorem 1 in this submission is a special case of Theorem 1 in submission 1613 (from the same authors). After some discussions, reviewers recognized that Theorem 2 is more important than Theorem 1, and is the main contribution of this work. Therefore, I recommend reject 1613 and accept this one. Regarding the stochastic variance-reduced EM, the following important reference is missing: [1] Zhu, Rongda, Lingxiao Wang, Chengxiang Zhai, and Quanquan Gu. "High-dimensional variance-reduced stochastic gradient expectation-maximization algorithm." In ICML, 2017.